# Linear-Time Sequence Modeling with Selective State Spaces

**Albert Gu**[*]
Machine Learning Department
Carnegie Mellon University
agu@cs.cmu.edu

**Tri Dao**[*]
Department of Computer Science
Princeton University
tri@tridao.me

## Abstract

Foundation models, now powering most of the exciting applications in deep learning, are almost universally based on the Transformer architecture and its core attention module. Many subquadratic-time architectures such as linear attention, gated convolution and recurrent models, and structured state space models (SSMs) have been developed to address Transformers' computational inefficiency on long sequences, but they have not performed as well as attention on important modalities such as language. We identify that a key weakness of such models is their inability to perform content-based reasoning, and make several improvements. First, simply letting the SSM parameters be functions of the input addresses their weakness with discrete modalities, allowing the model to *selectively* propagate or forget information along the sequence length dimension depending on the current token. Second, even though this change prevents the use of efficient convolutions, we design a hardware-aware parallel algorithm in recurrent mode. We integrate these selective SSMs into a simplified end-to-end neural network architecture without attention or even MLP blocks (**Mamba**). Mamba enjoys fast inference ($5\times$ higher throughput than Transformers) and linear scaling in sequence length, and its performance improves on real data up to million-length sequences. As a general sequence model backbone, Mamba achieves state-of-the-art performance across several modalities such as language, audio, and genomics. On language modeling, our Mamba-3B model outperforms Transformers of the same size and matches Transformers twice its size, both in pretraining and downstream evaluation.

## 1 Introduction

Foundation models (FMs), or large models pretrained on massive data then adapted for downstream tasks, have emerged as an effective paradigm in modern machine learning. The backbone of these FMs are often *sequence models*, operating on arbitrary sequences of inputs from a wide variety of domains such as language, images, speech, audio, time series, and genomics (Sutskever et al., 2014; Dosovitskiy et al., 2020; Oord et al., 2016; Brown et al., 2020; Ismail Fawaz et al., 2019; Poli et al., 2023). While this concept is agnostic to a particular choice of model architecture, modern FMs are predominantly based on a single type of sequence model: the Transformer (Vaswani et al., 2017) and its core attention layer. The efficacy of self-attention is attributed to its ability to route information densely within a context window, allowing it to model complex data. However, this property brings fundamental drawbacks: an inability to model anything outside of a finite window, and quadratic scaling with respect to the window length. An enormous body of research has appeared on more efficient variants of attention to overcome these drawbacks (Tay et al., 2022), but often at the expense of the very properties that makes it effective. As of yet, none of these variants have been shown to be empirically effective at scale across domains.

Recently, structured state space sequence models (SSMs) (Gu et al., 2021; 2022a) have emerged as a promising class of architectures for sequence modeling. These models can be interpreted as a combination of recurrent neural networks (RNNs) and convolutional neural networks (CNNs), with inspiration from classical state space models (Kalman, 1960). This class of models can be computed very efficiently as either a recurrence or convolution, with linear or near-linear scaling in sequence length. Additionally, they have principled mechanisms for modeling long-range dependencies (Gu et al., 2020a) in certain data modalities, and have dominated benchmarks such as the Long Range Arena (Tay et al., 2021). Many flavors of SSMs (Gu et al., 2022a; Gupta et al.,

---

[*]Alphabetical by first name.

2022a; Gu et al., 2022b; Li et al., 2023; Ma et al., 2023; Smith et al., 2023; Orvieto et al., 2023) have been successful in domains involving continuous signal data such as audio and vision (Goel et al., 2022; Saon et al., 2023; Nguyen et al., 2022). However, they have been less effective at modeling discrete and information-dense data such as text.

We propose a new class of **selective state space models**, that improves on prior work on several axes to achieve the modeling power of Transformers while scaling linearly in sequence length.

**Selection Mechanism.** First, we identify a key limitation of prior models: the ability to efficiently *select* data in an input-dependent manner (i.e. focus on or ignore particular inputs). Building on intuition based on important synthetic tasks such as selective copy and induction heads, we design a simple selection mechanism by parameterizing the SSM parameters based on the input. This allows the model to filter out irrelevant information and remember relevant information indefinitely.

**Hardware-aware Algorithm.** This simple change poses a technical challenge for the computation of the model; in fact, all prior SSMs models must be time- and input-invariant in order to be computationally efficient. We overcome this with a hardware-aware algorithm that computes the model recurrently with a scan instead of convolution, but does not materialize the expanded state in order to avoid IO access between different levels of the GPU memory hierarchy. The resulting implementation is faster than previous methods both in theory (scaling linearly in sequence length, compared to pseudo-linear for all convolution-based SSMs) and on modern hardware (up to $3\times$ faster on A100 GPUs).

**Architecture.** We simplify prior deep sequence model architectures by combining the design of prior SSM architectures (Dao et al., 2023) with the MLP block of Transformers into a single block, leading to a simple and homogenous architecture design (**Mamba**) incorporating selective state spaces.

Selective SSMs, and by extension the Mamba architecture, are fully recurrent models with key properties that make them suitable as the backbone of general foundation models operating on sequences. (i) High quality: selectivity brings strong performance on dense modalities such as language and genomics. (ii) Fast training and inference: computation and memory scales linearly in sequence length during training, and unrolling the model autoregressively during inference requires only constant time per step since it does not require a cache of previous elements. (iii) Long context: the quality and efficiency together yield performance improvements on real data up to sequence length 1M.

We empirically validate Mamba's potential as a general sequence FM backbone, in both pretraining quality and domain-specific task performance, on several types of modalities and settings:

- **Synthetics.** On important synthetic tasks such as copying and induction heads that have been proposed as being key to large language models, Mamba not only solves them easily but can *extrapolate solutions indefinitely long* ($>$1M tokens).

- **Audio and Genomics.** Mamba out-performs prior state-of-the-art models such as SaShiMi, Hyena, and Transformers on modeling audio waveforms and DNA sequences, both in pretraining quality and downstream metrics (e.g. reducing FID on a challenging speech generation dataset by more than half). In both settings, its *performance improves with longer context up to million-length sequences*.

- **Language Modeling.** Mamba is the first *linear-time sequence model that truly achieves Transformer-quality performance*, both in pretraining perplexity and downstream evaluations. With scaling laws up to 1B parameters, we show that Mamba exceeds the performance of a large range of baselines, including very strong modern Transformer training recipes based on LLaMa (Touvron et al., 2023). Our Mamba language model has $5\times$ generation throughput compared to Transformers of similar size, and Mamba-3B's quality matches that of Transformers twice its size (e.g. 4 points higher avg. on common sense reasoning compared to Pythia-3B and even exceeding Pythia-7B).

## 2  State Space Models

Structured state space sequence models (S4) are a recent class of sequence models for deep learning that are broadly related to RNNs, and CNNs, and classical state space models. They are inspired by a particular continuous system (1) that maps a 1-dimensional function or sequence $x(t) \in \mathbb{R} \mapsto y(t) \in \mathbb{R}$ through an implicit latent state $h(t) \in \mathbb{R}^N$. Concretely, S4 models are defined with four parameters $(\Delta, A, B, C)$, which define a sequence-to-sequence transformation in two stages.

$$h'(t) = \boldsymbol{A}h(t) + \boldsymbol{B}x(t) \quad \text{(1a)} \qquad h_t = \overline{\boldsymbol{A}}h_{t-1} + \overline{\boldsymbol{B}}x_t \quad \text{(2a)} \qquad \overline{K} = (\boldsymbol{C}\overline{\boldsymbol{B}}, \boldsymbol{C}\overline{\boldsymbol{A}}\overline{\boldsymbol{B}}, ..., \boldsymbol{C}\overline{\boldsymbol{A}}^k\overline{\boldsymbol{B}}, ...) \quad \text{(3a)}$$

$$y(t) = \boldsymbol{C}h(t) \quad \text{(1b)} \qquad y_t = \boldsymbol{C}h_t \quad \text{(2b)} \qquad y = x * \overline{K} \quad \text{(3b)}$$

**Discretization.** The first stage transforms the "continuous parameters" $(\Delta, \boldsymbol{A}, \boldsymbol{B})$ to "discrete parameters" $(\overline{\boldsymbol{A}}, \overline{\boldsymbol{B}})$ through fixed formulas $\overline{\boldsymbol{A}} = f_A(\Delta, \boldsymbol{A})$ and $\overline{\boldsymbol{B}} = f_B(\Delta, \boldsymbol{A}, \boldsymbol{B})$, where the pair $(f_A, f_B)$ is called a *discretization rule*. The most common is zero-order hold (ZOH) defined by $\overline{\boldsymbol{A}} = \exp(\Delta\boldsymbol{A})$ and $\overline{\boldsymbol{B}} = (\Delta\boldsymbol{A})^{-1}(\exp(\Delta\boldsymbol{A}) - \boldsymbol{I}) \cdot \Delta\boldsymbol{B}$.

Discretization has deep connections to continuous-time systems which can endow them with additional properties such as resolution invariance (Nguyen et al., 2022) and automatically ensuring that the model is properly normalized (Gu et al., 2023; Orvieto et al., 2023). It also has connections to gating mechanisms of RNNs (Tallec & Ollivier, 2018; Gu et al., 2020b) which we will revisit in Section 3.5. However, from a mechanical point of view discretization can simply be viewed as the first step of the computation graph in the forward pass of an SSM.

**Computation.** After the parameters have been transformed from $(\Delta, \boldsymbol{A}, \boldsymbol{B}, \boldsymbol{C}) \mapsto (\overline{\boldsymbol{A}}, \overline{\boldsymbol{B}}, \boldsymbol{C})$, the model can be computed in two ways, either as a **linear recurrence** (2) or a **global convolution** (3).

Commonly, the model uses the convolutional mode (3) for efficient parallelizable training (where the whole input sequence is seen ahead of time), and switched into recurrent mode (2) for efficient autoregressive inference (where the inputs are seen one timestep at a time).

**Linear Time Invariance (LTI).** An important property of equations (1) to (3) is that the model's dynamics are constant through time. In other words $(\Delta, \boldsymbol{A}, \boldsymbol{B}, \boldsymbol{C})$, and consequently $(\overline{\boldsymbol{A}}, \overline{\boldsymbol{B}})$ as well, are fixed for all time-steps. This property is called *linear time invariance (LTI)*, which is deeply connected to recurrence and convolutions. Informally, we think of LTI SSMs as being equivalent to any linear recurrence (2a) or convolution (3b), and use LTI as an umbrella term for these classes of models.

Thus far, all structured SSMs have been LTI (e.g. computed as convolutions) because of fundamental efficiency constraints, discussed in Section 3.3. However, a core insight of this work is that LTI models have fundamental limitations in modeling certain types of data, and our technical contributions involve removing the LTI constraint while overcoming the efficiency bottlenecks.

**Structure and Dimensions.** Finally, we note that structured SSMs are so named because computing them efficiently also requires imposing structure on the $\boldsymbol{A}$ matrix. The most popular form of structure is diagonal (Gupta et al., 2022a; Gu et al., 2022b; Smith et al., 2023), which we also use.

In this case, the $\boldsymbol{A} \in \mathbb{R}^{N \times N}, \boldsymbol{B} \in \mathbb{R}^{N \times 1}, \boldsymbol{C} \in \mathbb{R}^{1 \times N}$ matrices can all be represented by $N$ numbers. To operate over an input sequence $x$ of batch size $B$ and length $L$ with $D$ channels, the SSM is applied independently to each channel. Note that in this case, the total hidden state has dimension $DN$ per input, and computing it over the sequence length requires $O(BLDN)$ time and memory; this is the root of the fundamental efficiency bottleneck addressed in Section 3.3.

**SSM Architectures.** SSMs are standalone sequence transformations that can be incorporated into end-to-end neural network architectures. We discuss some of the most well-known SSM architectures, many of which will also serve as our primary baselines.

- Linear attention (Katharopoulos et al., 2020) is an approximation of self-attention involving a recurrence which can be viewed as a degenerate linear SSM.
- H3 (Dao et al., 2023) generalized this recurrence to use S4; it can be viewed as an architecture with an SSM sandwiched by two gated connections (Figure 2). H3 also inserts a standard local convolution, which they frame as a shift-SSM, before the main SSM layer.
- Hyena (Poli et al., 2023) uses the same architecture as H3 but replaces the S4 layer with an MLP-parameterized global convolution (Romero et al., 2021).

Other closely related SSMs and architectures are discussed further in an extended related work (Appendix B). We highlight in particular S5 (Smith et al., 2023), QRNN (Bradbury et al., 2016), and SRU (Lei et al., 2017), which we view as the most closely related methods to our core selective SSM.

## 3 Selective State Space Models

We motivate our selection mechanism using intuition from synthetic tasks (Section 3.1), then explain how to incorporate this mechanism into state space models (Section 3.2). The resulting time-varying SSMs cannot use convolutions, presenting a technical challenge of how to compute

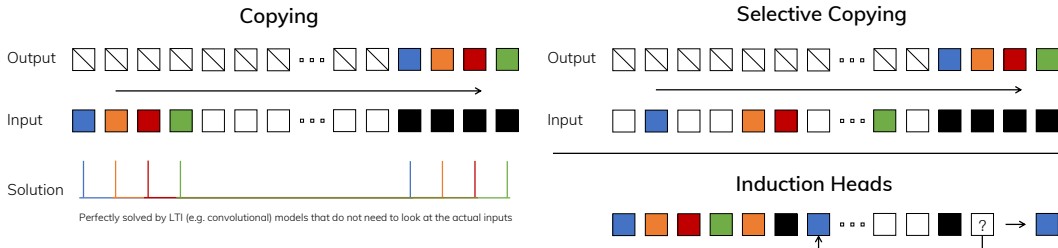

Figure 1: (*Left*) The standard version of the Copying task involves constant spacing between input and output elements and is easily solved by time-invariant models such as linear recurrences and global convolutions. (*Right Top*) The Selective Copying task has random spacing in between inputs and requires time-varying models that can *selectively* remember or ignore inputs depending on their content. (*Right Bottom*) The Induction Heads task is an example of associative recall that requires retrieving an answer based on context, a key ability for LLMs.

them efficiently. We overcome this with a hardware-aware algorithm that exploits the memory hierarchy on modern hardware (Section 3.3). We then describe a simple SSM architecture without attention or even MLP blocks (Section 3.4). Finally, we discuss some additional properties of selection mechanisms (Section 3.5).

### 3.1 Motivation: Selection as a Means of Compression

We argue that a fundamental problem of sequence modeling is *compressing context into a smaller state*. In fact, we can view the tradeoffs of popular sequence models from this point of view. For example, attention is both effective and inefficient because it explicitly does not compress context at all. This can be seen from the fact that autoregressive inference requires explicitly storing the entire context (i.e. the KV cache), which directly causes the slow linear-time inference and quadratic-time training of Transformers. On the other hand, recurrent models are efficient because they have a finite state, implying constant-time inference and linear-time training. However, their effectiveness is limited by how well this state has compressed the context.

To understand this principle, we focus on two running examples of synthetic tasks (Figure 1).

- The **Selective Copying** task modifies the popular Copying task (Arjovsky et al., 2016) by varying the position of the tokens to memorize. It requires *content-aware* reasoning to be able to memorize the relevant tokens (*colored*) and filter out the irrelevant ones (*white*).

- The **Induction Heads** task is a well-known mechanism hypothesized to explain the majority of in-context learning abilities of LLMs (Olsson et al., 2022). It requires *context-aware* reasoning to know when to produce the correct output in the appropriate context (*black*).

These tasks reveal the failure mode of LTI models. From the recurrent view, their constant dynamics (e.g. the $(\overline{A},\overline{B})$ transitions in (2)) cannot let them select the correct information from their context, or affect the hidden state passed along the sequence in an input-dependent way. From the convolutional view, it is known that global convolutions can solve the vanilla Copying task (Romero et al., 2021) because it only requires time-awareness, but that they have difficulty with the Selective Copying task because of lack of content-awareness (Figure 1). More concretely, the spacing between inputs-to-outputs is varying and cannot be modeled by static convolution kernels.

In summary, the efficiency vs. effectiveness tradeoff of sequence models is characterized by how well they compress their state: efficient models must have a small state, while effective models must have a state that contains all necessary information from the context. In turn, we propose that a fundamental principle for building sequence models is **selectivity**: or the context-aware ability to focus on or filter out inputs into a sequential state. In particular, a selection mechanism controls how information propagates or interacts along the sequence dimension (see Section 3.5 for more discussion).

### 3.2 Improving SSMs with Selection

One method of incorporating a selection mechanism into models is by letting their parameters that affect interactions along the sequence (e.g. the recurrent dynamics of an RNN or the convolution kernel of a CNN) be input-dependent.

Algorithms 1 and 2 illustrates the main selection mechanism that we use. The main difference is simply making several parameters $\Delta, B, C$ functions of the input, along with the associated changes to tensor shapes throughout. In particular, we highlight that these parameters now have a length dimension $L$, meaning that the model has changed from time-invariant to time-varying. (Note

| **Algorithm 1** SSM (S4) | **Algorithm 2** SSM + Selection (S6) |
|---|---|
| **Input:** $x$ : (B,L,D) | **Input:** $x$ : (B,L,D) |
| **Output:** $y$ : (B,L,D) | **Output:** $y$ : (B,L,D) |
| 1: $A$ : (D,N) ← Parameter | 1: $A$ : (D,N) ← Parameter |
|    ▷ Represents structured $N \times N$ matrix |    ▷ Represents structured $N \times N$ matrix |
| 2: $B$ : (D,N) ← Parameter | 2: $B$ : (B,L,N) ← $s_B(x)$ |
| 3: $C$ : (D,N) ← Parameter | 3: $C$ : (B,L,N) ← $s_C(x)$ |
| 4: $\Delta$ : (D) ← $\tau_\Delta$ (Parameter) | 4: $\Delta$ : (B,L,D) ← $\tau_\Delta$ (Parameter $+ s_\Delta(x)$) |
| 5: $\overline{A}, \overline{B}$ : (D,N) ← discretize$(\Delta, A, B)$ | 5: $\overline{A}, \overline{B}$ : (B,L,D,N) ← discretize$(\Delta, A, B)$ |
| 6: $y \leftarrow$ SSM$(\overline{A}, \overline{B}, C)(x)$ | 6: $y \leftarrow$ SSM$(\overline{A}, \overline{B}, C)(x)$ |
|    ▷ Time-invariant: recurrence or convolution |    ▷ Time-varying: recurrence (*scan*) only |
| 7: **return** $y$ | 7: **return** $y$ |

that shape annotations were described in Section 2.) This loses the equivalence to convolutions (3) with implications for its efficiency, discussed next.

We specifically choose $s_B(x) = \text{Linear}_N(x)$, $s_C(x) = \text{Linear}_N(x)$, $s_\Delta(x) = \text{Broadcast}_D(\text{Linear}_1(x))$, and $\tau_\Delta = \text{softplus}$, where $\text{Linear}_d$ is a parameterized projection to dimension $d$. The choice of $s_\Delta$ and $\tau_\Delta$ is due to a connection to RNN gating mechanisms explained in Section 3.5.

### 3.3 Efficient Implementation of Selective SSMs

Hardware-friendly primitives such as convolutions (Krizhevsky et al., 2012) and attention (Bahdanau et al., 2015; Vaswani et al., 2017) enjoy widespread application. Here we aim to make selective SSMs efficient on modern hardware (GPUs) as well. The selection mechanism is quite natural, and earlier works attempted to incorporate special cases of selection, such as letting $\Delta$ vary over time in recurrent SSMs (Gu et al., 2020a). However, this was computationally difficult, which was why S4 and all derivatives used LTI (non-selective) models, most commonly in the form of global convolutions.

The selection mechanism is designed to overcome the limitations of LTI models; at the same time, we therefore need to revisit the computation problem of SSMs. We address this with three classical techniques: kernel fusion, parallel scan, and recomputation. We make two main observations:

- The naive recurrent computation uses $O(BLDN)$ FLOPs while the convolutional computation uses $O(BLD\log(L))$ FLOPs, and the former has a lower constant factor. Thus for long sequences and not-too-large state dimension $N$, the recurrent mode can actually use fewer FLOPs.
- The two challenges are the sequential nature of recurrence, and the large memory usage. To address the latter, just like the convolutional mode, we can attempt to not actually materialize the full state $h$.

The main idea is to leverage properties of modern accelerators (GPUs) to materialize the state $h$ only in more efficient levels of the memory hierarchy. In particular, most operations (except matrix multiplication) are bounded by memory bandwidth (Williams et al., 2009; Ivanov et al., 2021; Dao et al., 2022). This includes our scan operation, and we use kernel fusion to reduce the amount of memory IOs, leading to a significant speedup compared to a standard implementation.

Concretely, instead of preparing the scan input $(\overline{A}, \overline{B})$ of size (B,L,D,N) in GPU HBM (high-bandwidth memory), we load the SSM parameters $(\Delta, A, B, C)$ directly from slow HBM to fast SRAM, perform the discretization and recurrence in SRAM, and then write the final outputs of size (B,L,D) back to HBM.

To avoid the sequential recurrence, we observe that despite not being linear it can still be parallelized with a work-efficient parallel scan algorithm (Blelloch, 1990; Martin & Cundy, 2018; Smith et al., 2023).

Details of the fused kernel and recomputation are in Appendix D.

### 3.4 A Simplified SSM Architecture

As with structured SSMs, selective SSMs are standalone sequence transformations that can be flexibly incorporated into neural networks. The H3 architecture is the basis for the most well-known SSM architectures (Section 2), which are generally comprised of a block inspired by linear attention interleaved with an MLP (multi-layer perceptron) block. We simplify this architecture by combining these two components into one, which is stacked homogenously (Figure 2).

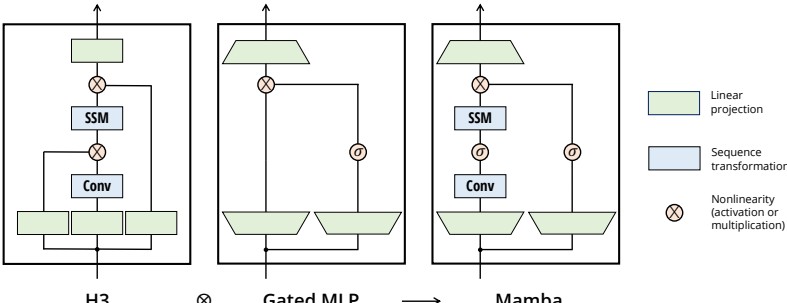

Figure 2: (**Architecture**.) Our simplified block design combines the H3 block, which is the basis of most SSM architectures, with the ubiquitous MLP block of modern neural networks. Instead of interleaving these two blocks, we simply repeat the Mamba block homogenously. Compared to the H3 block, Mamba replaces the first multiplicative gate with an activation function. Compared to the MLP block, Mamba adds an SSM to the main branch. For $\sigma$ we use the SiLU / Swish activation (Hendrycks & Gimpel, 2016; Ramachandran et al., 2017).

This architecture involves expanding the model dimension $D$ by a controllable expansion factor $E$. For each block, most of the parameters ($3ED^2$) are in the linear projections while the inner SSM contributes less. We always fix to $E=2$ in our experiments and use two stacks of the block to match the $12D^2$ parameters of a Transformer's interleaved MHA (multi-head attention) and MLP blocks.

### 3.5 Properties of Selection Mechanisms

The selection mechanism is a broader concept that can be applied in different ways, such as to more traditional RNNs or CNNs, to different parameters (e.g. $A$ in Algorithm 2), or using different transformations $s(x)$.

We highlight the most important connection: the classical gating mechanism of RNNs is an instance of our selection mechanism for SSMs. We note that the connection between RNN gating and the discretization of continuous-time systems is well established (Funahashi & Nakamura, 1993; Tallec & Ollivier, 2018). In fact, Theorem 1 is an improvement of Gu et al. (2021, Lemma 3.1) generalizing to the ZOH discretization and input-dependent gates (proof in Appendix C). More broadly, $\Delta$ in SSMs can be seen to play a generalized role of the RNN gating mechanism. In line with prior work, we adopt the view that *discretization of SSMs is the principled foundation of heuristic gating mechanisms*.

**Theorem 1.** *When $N=1, A=-1, B=1, s_\Delta = \text{Linear}(x)$, and $\tau_\Delta = \text{softplus}$, then the selective SSM recurrence (Algorithm 2) takes the form* $g_t = \sigma(\text{Linear}(x_t))$ *(the* gate*) and* $h_t = (1-g_t)h_{t-1} + g_t x_t$.

As mentioned in Section 3.2, our specific choices of $s_\Delta, \tau_\Delta$ is from this connection. In particular, note that if a given input $x_t$ should be completely ignored (as necessary in the synthetic tasks), all $D$ channels should ignore it, and so we project the input down to 1 dimension before repeating/broadcasting with $\Delta$.

We elaborate on two particular mechanistic effects of selection.

**Variable Spacing.** Selectivity allows filtering out irrelevant noise tokens that may occur between inputs of interest. This is exemplified by the Selective Copying task, but occurs ubiquitously in common data modalities, particularly for discrete data – for example the presence of language fillers such as "um". This property arises because the model can mechanistically filter out any particular input $x_t$, for example in the gated RNN case (Theorem 1) when $g_t \to 0$.

**Filtering Context.** It has been empirically observed that many sequence models do not improve with longer context (Shi et al., 2023a), despite the principle that more context should lead to strictly better performance. An explanation is that many sequence models cannot effectively ignore irrelevant context when necessary; an intuitive example are global convolutions (and general LTI models). On the other hand, selective models can simply reset their state at any time to remove extraneous history, and thus their performance in principle improves monotonically with context length (e.g. Appendix E.2.2).

We remark that while the $A$ parameter could also be selective, it ultimately affects the model only through its interaction with $\Delta$ via $\overline{A} = \exp(\Delta A)$ (the discretization. Thus selectivity in $\Delta$ is enough to ensure selectivity in $(\overline{A}, \overline{B})$, and is the main source of improvement. We hypothesize that making $A$ selective in addition to (or instead of) $\Delta$ would have similar performance, and leave it out for simplicity.

| MODEL | ARCH. | LAYER | ACC. |
|-------|-------|-------|------|
| S4 | No gate | S4 | 18.3 |
| - | No gate | S6 | **97.0** |
| H3 | H3 | S4 | 57.0 |
| Hyena | H3 | Hyena | 30.1 |
| - | H3 | S6 | **99.7** |
| - | Mamba | S4 | 56.4 |
| - | Mamba | Hyena | 28.4 |
| Mamba | Mamba | S6 | **99.8** |

Table 1: (**Selective Copying**.) Accuracy for combinations of architectures and inner sequence layers.

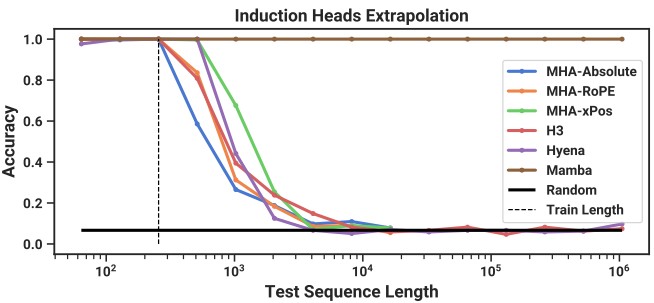

Figure 3: (**Induction Heads**.) Models are trained on sequence length $2^8 = 256$, and tested on increasing sequence lengths of $2^6 = 64$ up to $2^{20} = 1048576$. Full numbers in Table 10.

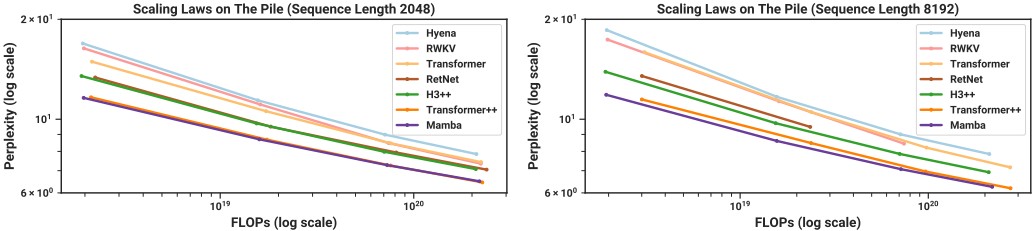

Figure 4: (**Scaling Laws**.) Models of size $\approx 125M$ to $\approx 1.3B$ parameters, trained on the Pile. Mamba scales better than all other attention-free models and is the first to match the performance of a very strong "Transformer++" recipe that has now become standard, particularly as the sequence length grows.

**Remark 3.1.** *For brevity in our experimental results, we sometimes abbreviate selective SSMs as* S6 *models, because they are S4 models with a* selection *mechanism and computed with a* scan.

## 4 Empirical Evaluation

Mamba achieves state-of-the-art results on the synthetic tasks (Section 4.1) and three different domains (language, DNA, audio) (Sections 4.2 to 4.4) on both pretraining and downstream tasks, while being very computationally efficient (Appendix E.4).

### 4.1 Synthetic Tasks

Table 1 and Figure 3 show results for the synthetic tasks. On Selective Copying, the selective SSM layer is enough to solve the task independently of the architecture used, while previous LTI SSMs cannot even when combined with more powerful architectures. On Induction Heads, Mamba learns the task perfectly and can even extrapolate to million-length sequences, or $4000\times$ longer than it saw during training, while no other method goes beyond $2\times$. Full discussion for synthetic tasks are in Appendix E.1.

### 4.2 Language Modeling

We evaluate the Mamba architecture on standard autoregressive language modeling against other architectures, on both pretraining metrics (perplexity) and zero-shot evaluations. We set the model sizes (depth and width) to mirror GPT3 specifications. We use the Pile dataset (Gao et al., 2020), and follow the training recipe described in Brown et al. (2020). All training details are in Appendix F.2.

**Scaling Laws.** For baselines, we compare against the standard Transformer architecture (GPT3 architecture), as well as the strongest Transformer recipe we know of (here referred to as Transformer++), based on the PaLM and LLaMa architectures (e.g. rotary embedding, SwiGLU MLP, etc.). We also compare against other recent subquadratic architectures (Figure 4). All model details are in Appendix F.2.

**Downstream Evaluations.** Table 2 shows the performance of Mamba on a range of popular downstream zero-shot evaluation tasks. We compare against the most well-known open source models at these sizes, most importantly Pythia (Biderman et al., 2023) and RWKV (Peng et al., 2023) which were trained with the same tokenizer, dataset, and training length (300B tokens) as our models.

Table 2: (**Zero-shot Evaluations**.) Best results for each size in bold. We compare against open source LMs with various tokenizers, trained for up to 300B tokens. Pile refers to the validation split, comparing only against models trained on the same dataset and tokenizer (GPT-NeoX-20B). For each model size, Mamba is best-in-class on every single evaluation result, and generally matches baselines at twice the model size.

| MODEL | TOKEN. | PILE PPL ↓ | LAMBADA PPL ↓ | LAMBADA ACC ↑ | HELLASWAG ACC ↑ | PIQA ACC ↑ | ARC-E ACC ↑ | ARC-C ACC ↑ | WINOGRANDE ACC ↑ | AVERAGE ACC ↑ |
|---|---|---|---|---|---|---|---|---|---|---|
| Hybrid H3-360M | GPT2 | — | 12.58 | 48.0 | 41.5 | 68.1 | 51.4 | 24.7 | 54.1 | 48.0 |
| Pythia-410M | NeoX | 9.95 | 10.84 | 51.4 | 40.6 | 66.9 | 52.1 | 24.6 | 53.8 | 48.2 |
| **Mamba-370M** | NeoX | **8.28** | **8.14** | **55.6** | **46.5** | **69.5** | **55.1** | **28.0** | **55.3** | **50.0** |
| Pythia-1B | NeoX | 7.82 | 7.92 | 56.1 | 47.2 | 70.7 | 57.0 | 27.1 | 53.5 | 51.9 |
| **Mamba-790M** | NeoX | **7.33** | **6.02** | **62.7** | **55.1** | **72.1** | **61.2** | **29.5** | **56.1** | **57.1** |
| GPT-Neo 1.3B | GPT2 | — | 7.50 | 57.2 | 48.9 | 71.1 | 56.2 | 25.9 | 54.9 | 52.4 |
| Hybrid H3-1.3B | GPT2 | — | 11.25 | 49.6 | 52.6 | 71.3 | 59.2 | 28.1 | 56.9 | 53.0 |
| OPT-1.3B | OPT | — | 6.64 | 58.0 | 53.7 | 72.4 | 56.7 | 29.6 | 59.5 | 55.0 |
| Pythia-1.4B | NeoX | 7.51 | 6.08 | 61.7 | 52.1 | 71.0 | 60.5 | 28.5 | 57.2 | 55.2 |
| RWKV-1.5B | NeoX | 7.70 | 7.04 | 56.4 | 52.5 | 72.4 | 60.5 | 29.4 | 54.6 | 54.3 |
| **Mamba-1.4B** | NeoX | **6.80** | **5.04** | **64.9** | **59.1** | **74.2** | **65.5** | **32.8** | **61.5** | **59.7** |
| GPT-Neo 2.7B | GPT2 | — | 5.63 | 62.2 | 55.8 | 72.1 | 61.1 | 30.2 | 57.6 | 56.5 |
| Hybrid H3-2.7B | GPT2 | — | 7.92 | 55.7 | 59.7 | 73.3 | 65.6 | 32.3 | 61.4 | 58.0 |
| OPT-2.7B | OPT | — | 5.12 | 63.6 | 60.6 | 74.8 | 60.8 | 31.3 | 61.0 | 58.7 |
| Pythia-2.8B | NeoX | 6.73 | 5.04 | 64.7 | 59.3 | 74.0 | 64.1 | 32.9 | 59.7 | 59.1 |
| RWKV-3B | NeoX | 7.00 | 5.24 | 63.9 | 59.6 | 73.7 | 67.8 | 33.1 | 59.6 | 59.6 |
| **Mamba-2.8B** | NeoX | **6.22** | **4.23** | **69.2** | **66.1** | **75.2** | **69.7** | **36.3** | **63.5** | **63.3** |
| GPT-J-6B | GPT2 | – | 4.10 | 68.3 | 66.3 | 75.4 | 67.0 | 36.6 | 64.1 | 63.0 |
| OPT-6.7B | OPT | – | 4.25 | 67.7 | 67.2 | 76.3 | 65.6 | 34.9 | 65.5 | 62.9 |
| Pythia-6.9B | NeoX | 6.51 | 4.45 | 67.1 | 64.0 | 75.2 | 67.3 | 35.5 | 61.3 | 61.7 |
| RWKV-7.4B | NeoX | 6.31 | 4.38 | 67.2 | 65.5 | 76.1 | 67.8 | 37.5 | 61.0 | 62.5 |

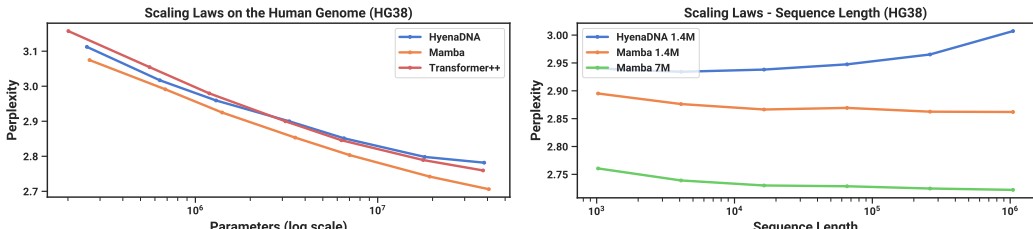

Figure 5: (**DNA Scaling Laws**.) Pretraining on the HG38 (human genome) dataset. (*Left*) Fixing short context length $2^{10} = 1024$ and increasing size from $\approx 200K$ to $\approx 40M$ parameters, Mamba scales better than baselines. (*Right*) Fixing model size and increasing sequence lengths while controlling for computation. Unlike baselines, the selection mechanism of Mamba facilitates better performance with increasing context length.

## 4.3 DNA Modeling

Motivated by the success of large language models, there has been recent exploration into using the foundation model paradigm for genomics. DNA has been likened to language in that it consists of sequences of discrete tokens with a finite vocabulary. It is also known for requiring long-range dependencies to model (Avsec et al., 2021). We investigate Mamba as a FM backbone for pretraining and fine-tuning in the same setting as recent works on long-sequence models for DNA (Nguyen et al., 2023). In particular, we focus on two explorations of scaling laws across model size and sequence length (Figure 5), and a difficult downstream synthetic classification task requiring long context (Figure 6). Full discussion for these tasks is in Appendix E.2.

## 4.4 Audio Modeling and Generation

For the audio waveform modality, we compare primarily to the SaShiMi architecture and training protocols (Goel et al., 2022). The architecture is a U-Net with alternating S4 and MLP blocks, which we consider replacing with Mamba. Experiment details are in Appendix F.4.

**Long-Context Autoregressive Pretraining.** We evaluate pretraining quality on YouTubeMix (Deep-Sound, 2017), a standard piano music dataset. Figure 7 evaluates the effect of increasing training sequence lengths from $2^{13} = 8192$ to $2^{20} \approx 10^6$, while keeping computation fixed.

**Autoregressive Speech Generation.** Tables 3 and 4 show results on the SC09 dataset for generating speech clips of the digits 0-9 (Warden, 2018; Donahue et al., 2019). Mamba sets significant state-of-the-art results, and is consistently better than attention or S4 blocks in a parameter-controlled setting.

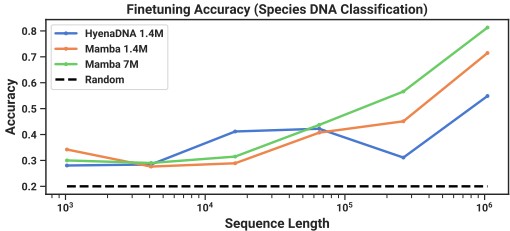

Figure 6: (**Great Apes DNA Classification**.) Accuracy after fine-tuning on sequences of length $2^{10} = 1024$ up to $2^{20} = 1048576$ using pretrained models of the same context length. Numerical results in Table 12.

Figure 7: (**Audio Pretraining**.) Mamba improves performance over prior state-of-the-art (Sashimi) in autoregressive audio modeling, while improving up to minute-long context or million-length sequences.

Table 3: (**SC09**) Automated metrics for unconditional generation on a challenging dataset of fixed-length speech clips. (*Top to Bottom*) Autoregressive baselines, non-autoregressive baselines, Mamba, and dataset metrics.

| MODEL | PARAMS | NLL ↓ | FID ↓ | IS ↑ | mIS ↑ | AM ↓ |
|---|---|---|---|---|---|---|
| SampleRNN | 35.0M | 2.042 | 8.96 | 1.71 | 3.02 | 1.76 |
| WaveNet | 4.2M | 1.925 | 5.08 | 2.27 | 5.80 | 1.47 |
| SaShiMi | 5.8M | 1.873 | 1.99 | 5.13 | 42.57 | 0.74 |
| WaveGAN | 19.1M | - | 2.03 | 4.90 | 36.10 | 0.80 |
| DiffWave | 24.1M | - | 1.92 | 5.26 | 51.21 | 0.68 |
| + SaShiMi | 23.0M | - | 1.42 | 5.94 | 69.17 | 0.59 |
| **Mamba** | 6.1M | **1.852** | 0.94 | 6.26 | 88.54 | 0.52 |
| **Mamba** | 24.3M | 1.860 | **0.67** | 7.33 | 144.9 | **0.36** |
| Train | - | - | 0.00 | 8.56 | 292.5 | 0.16 |
| Test | - | - | 0.02 | 8.33 | 257.6 | 0.19 |

Table 4: (**SC09 Model Ablations**) Models with 6M parameters. In SaShiMi's U-Net backbone, there are 8 center blocks operating on sequence length 1000, sandwiched on each side by 8 outer blocks on sequence length 4000, sandwiched by 8 outer blocks on sequence length 16000 (40 blocks total). The architecture of the 8 center blocks are ablated independently of the rest. Note that Transformers (MHA+MLP) were not tested in the more important outer blocks because of efficiency constraints.

| OUTER | CENTER | NLL ↓ | FID ↓ | IS ↑ | mIS ↑ | AM ↓ |
|---|---|---|---|---|---|---|
| S4+MLP | MHA+MLP | 1.859 | 1.45 | 5.06 | 47.03 | 0.70 |
| S4+MLP | S4+MLP | 1.867 | 1.43 | 5.42 | 53.54 | 0.65 |
| S4+MLP | Mamba | 1.859 | 1.42 | 5.71 | 56.51 | 0.64 |
| Mamba | MHA+MLP | **1.850** | 1.37 | 5.63 | 58.23 | 0.62 |
| Mamba | S4+MLP | 1.853 | 1.07 | 6.05 | 73.34 | 0.55 |
| Mamba | Mamba | 1.852 | **0.94** | **6.26** | **88.54** | **0.52** |

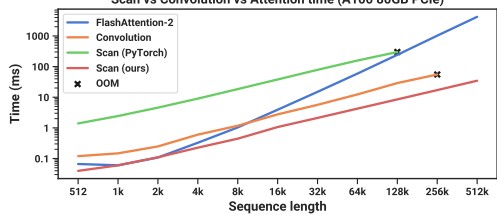

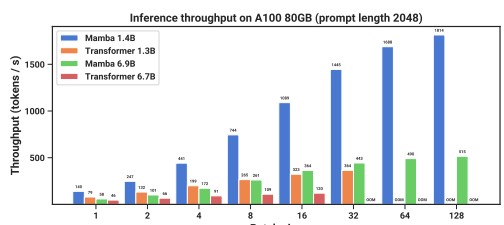

Figure 8: (**Efficiency Benchmarks**.) (*Left*) Training: our efficient scan is 40× faster than a standard implementation. (*Right*) Inference: as a recurrent model, Mamba can achieve 5× higher throughput than Transformers.

## 5  Conclusion

We introduce a selection mechanism to structured state space models, allowing them to perform context-dependent reasoning while scaling linearly in sequence length. When incorporated into a simple attention-free architecture, Mamba achieves state-of-the-art results on a diverse set of domains, where it matches or exceeds the performance of strong Transformer models. We are excited about the broad applications of selective state space models to build foundation models for different domains, especially in emerging modalities requiring long context such as genomics, audio, and video. Our results suggest that Mamba is a strong candidate to be a general sequence model backbone.

## Acknowledgments

We thank Karan Goel, Arjun Desai, and Kush Bhatia for helpful feedback on the draft.

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

## A  Discussion: Selection Mechanism

Our selection mechanism is inspired by and related to concepts such as gating, hypernetworks, and data-dependence. It can also be viewed as related to "fast weights" (Schmidhuber, 1992; Ba et al., 2016), which connects classical RNNs with the mechanism of linear attention (Schlag et al., 2021). However, we believe that it is a distinct concept that is worth clarifying.

**Gating.**  Gating originally referred to the gating mechanisms of RNNs such as the LSTM (Hochreiter & Schmidhuber, 1997) and GRU (Chung et al., 2014), or the gated equation in Theorem 1. This was interpreted as a particular mechanism for controlling whether to let an input into the hidden state of an RNN. In particular, this affects the propagation of signal through time and causes inputs to interact along the sequence length dimension.

However, the concept of gating has since been relaxed in popular usage to simply mean any multiplicative interaction (often with an activation function). For example, *elementwise* multiplicative components of neural network architectures (that do not interact along sequence length) are now commonly referred to as gated architectures (Hua et al., 2022; Mehta et al., 2023), despite a very different meaning than the original RNN sense. Thus we believe the original concept of *RNN gating* versus the popular usage of *multiplicative gating* actually have a very different semantic meaning.

**Hypernetworks.**  Hypernetworks refer to neural networks whose parameters are themselves generated by smaller neural networks. The original idea (Ha et al., 2017) used it in a narrow sense to define a large RNN whose recurrent parameters are generated by a smaller RNN, and other variants have been around for a long time (Schmidhuber, 1992).

**Data-dependence.**  Similar to hypernetworks, data-dependence can refer to any notion where some parameters of the model depend on the data (Poli et al., 2023).

**Example: GLU Activation.**  To illustrate the issues with these concepts, consider a simple diagonal linear layer $y = Dx$, where $D$ is a diagonal weight parameter. Now suppose that $D$ is itself generated from a linear transformation of $x$, with an optional nonlinearity: $D = \sigma(Wx)$. Since it is diagonal, the multiplication becomes an elementwise product: $y = \sigma(Wx) \circ x$.

This is a rather trivial transformation, yet it technically satisfies the common meanings of gating (since it has a multiplicative "branch"), hypernetworks (since the parameter $D$ is generated by another layer), and data-dependent (since $D$ depends on the data $x$). However, this in fact simply defines a GLU function, which is so simple that it is often considered just an activation function (Dauphin et al., 2017; Shazeer, 2020) instead of a meaningful layer.

**Selection.**  Thus, while selection mechanisms could be considered a special case of ideas such as architectural gating, hypernetworks, or data-dependence, so can an enormous range of other constructions—essentially anything with a multiplication, including standard attention mechanisms (Bahdanau et al., 2015; Vaswani et al., 2017) as well—and we find it uninformative to think of them as such.

Instead, we view it as most closely related to the gating mechanism of traditional RNNs, which is a special case (Theorem 1) and also has a deeper history of connections to SSMs through variable (input-dependent) discretization of $\Delta$ (Funahashi & Nakamura, 1993; Tallec & Ollivier, 2018; Gu et al., 2020a). We also eschew the term "gating" in favor of *selection* to clarify the overloaded use of former. More narrowly, we use selection to refer to the *mechanistic* action of a model to select or ignore inputs and facilitate data interaction along the sequence length (Section 3.1). Beyond selective SSMs and gated RNNs, other examples may include input-dependent convolutions (Yang et al., 2019; Lioutas & Guo, 2020; Kosma et al., 2023; Lutati et al., 2023) and even attention.

## B  Related Work

We overview several prior works related to our methods. We mention that some of the most closely related models include recurrent layers such as S4, S5, and quasi-RNNs; as well as end-to-end architectures such as H3, RetNet, and RWKV.

### B.1  S4 Variants and Derivatives

We describe a brief overview of some structured SSMs from past work, particularly those that have a relation to our method.

- S4 (Gu et al., 2021; 2022a) introduced the first structured SSM, describing diagonal structure and diagonal plus low-rank (DPLR). It focused on efficient convolutional algorithms for DPLR SSMs due to a connection to continuous-time online memorization (HIPPO) (Gu et al., 2020a).

- DSS (Gupta et al., 2022a) first discovered the empirical effectiveness of diagonal structured SSMs by approximating the HIPPO initialization. This was expanded on theoretically in S4D (Gu et al., 2022b).

- S5 (Smith et al., 2023) independently discovered the diagonal SSM approximation, and is the first S4 model to be computed recurrently with the parallel scan. However, this required lowering the effective state dimension, which they accomplished by switching the SSM dimensions from a SISO (single-input single-output) to MIMO (multi-input multi-output) formulation. Our proposed S6 shares the scan, but differs by (i) keeping the SISO dimensions, which provides a larger effective recurrent state, (ii) using a hardware-aware algorithm to overcome the computation issue, (iii) adding the selection mechanism.

  Lu et al. (2023) applied S5 to meta-RL in order to handle resetting the SSM state between episode trajectories. Their mechanism can be viewed as a particular hard-coded instance of a selection mechanism, where $\overline{A}$ is manually set to 0, instead of our learnable mechanism that depends on the input. It would be interesting to apply selective SSMs generically to this setting and probe if the model has learned to automatically reset its state on episode boundaries.

- Mega (Ma et al., 2023) introduced a simplification of S4 to be real- instead of complex- valued, giving it an interpretation of being an exponential moving average (EMA). They additionally make an interesting connection of the discretization step of SSMs to an EMA *damping* term. Contrary to findings in the original S4 papers, this was the first model to show that real-valued SSMs are empirically effective in certain settings or when combined with different architectural components.

- Liquid S4 (Hasani et al., 2023) is also motivated by augmenting S4 with an input-dependent state transition. From this perspective it shares similarity to selection mechanisms, although in a limited form which is still computed convolutionally and close to LTI.

- SGConv (Li et al., 2023), Hyena (Poli et al., 2023), LongConv (Fu et al., 2023), MultiresConv (Shi et al., 2023b), and Toeplitz Neural Network (Qin et al., 2023) all focus on the convolutional representation of S4 and create global or long convolution kernels with different parameterizations. However, these methods cannot do fast autoregressive inference directly.

Notably, all of these methods, and all other structured SSMs that we are aware of, have been non-selective and usually strictly LTI (linear time invariant).

## B.2 SSM Architectures

We use SSM architectures or state space neural networks (SSNN) to refer to deep neural network architectures incorporating one of the previous SSMs as a black box layer.

- GSS (Mehta et al., 2023) was the first gated neural network architecture incorporating SSMs. It is motivated by the gated attention unit (GAU) of Hua et al. (2022) and looks quite similar to our block, except with additional projections. Most importantly, its projection *contracts* the model dimension to reduce the state size of the SSM, while ours *expands* the model dimension in order to increase the state size, based on the motivation in Section 3.1.

- Mega (Ma et al., 2023) combined the EMA simplification of S4 described above into a hybrid architecture using an efficient attention approximation.

- H3 (Dao et al., 2023) is motivated by combining S4 with linear attention (Katharopoulos et al., 2020). It is the first to generalize this formulation of linear attention to more general recurrences, which is also the basis of later architectures.

- Selective S4 (Wang et al., 2023) incorporates S4 as a black box to generate a binary mask which is multiplied on the input. While sharing the "selection" name, we consider this an architectural modification that is closer to architectural gating than a selection mechanism (Appendix A). For example, we hypothesize that it would not solve the Selective Copying task because simply masking out the irrelevant inputs does not affect the spacing between the relevant ones (indeed, the Selective Copying task can even be viewed as coming pre-masked if the noise tokens are embedded to 0).

- RetNet (Sun et al., 2023) is also based on Linear Attention and very similar to H3, but reduces the inner S4 layer to a special case where the state dimension is $N = 1$. Although not framed as such, its recurrence can be viewed as a special case of a linear SSM.

Its primary source of improvement is using a linear attention with large *head dimension*, which can be viewed as another method to perform input-dependent state expansion. Using a larger head dimension in the context of linear attention variants was first done by H3, but not extensively used since this requires a proportional amount of extra computation. RetNet avoids this with an alternate way to parallelize the computation with a variant of standard multi-head attention instead of convolutions, made feasible by their particular special case of SSMs which acts as a simple EMA.

- RWKV (Peng et al., 2023) is another recent RNN designed for language modeling. It is based on AFT (attention-free Transformer (Zhai et al., 2021)), another variant of linear attention. Its main "WKV" mechanism involves LTI recurrences and can be seen as the ratio of two SSMs.

We also highlight the gated attention unit (GAU) from Hua et al. (2022), which was motivated by combining the Transformer's MHA and MLP blocks together and was an inspiration for our architecture (Section 3.4) combining the H3 and MLP blocks.

### B.3 Relationship to RNNs

RNNs and SSMs are broadly related, as they both involve the concepts of *recurrence* on a latent *state*.

Several older RNNs such as the strongly typed RNN (Balduzzi & Ghifary, 2016), quasi-RNN (QRNN) (Bradbury et al., 2016), and simple recurrent unit (SRU) (Lei et al., 2017; Lei, 2021) involve forms of gated RNNs without time-wise nonlinearities. Because of the connections of gating mechanisms and selection mechanisms, these can be viewed as cases of selective SSMs, and are thus more powerful in a sense than the family of LTI structured SSMs above. The main differences are:

- They do not use state expansion ($N=1$) or selective $B,C$ parameters, both of which are important for performance (Appendix E.3).

- They use a heuristic gating mechanism, which we generalize as a consequence of the selection mechanism + discretization (Theorem 1). The connections to principled SSM theory provides better parameterizations and initializations.

Additionally, older RNNs famously suffered from efficiency issues and the vanishing gradients problem (Hochreiter, 1991; Hochreiter et al., 2001; Pascanu et al., 2013), both caused by their sequential nature. The former could be solved for some of the above RNNs by leveraging the parallel scan (Martin & Cundy, 2018), but the latter was difficult without theory later developed for SSMs. For example, modern structured SSMs differ in more careful parameterization of the recurrent dynamics inspired by classical SSM theory (e.g. through discretization (Gu et al., 2021; 2023)), or direct analysis (Orvieto et al., 2023; Kaul, 2020; Gupta et al., 2022b)).

We also note that there is a long line of work on orthogonal RNNs (Arjovsky et al., 2016; Henaff et al., 2016; Mhammedi et al., 2017; Vorontsov et al., 2017; Lezcano-Casado & Martínez-Rubio, 2019) which are motivated by constraining the $\overline{A}$ transition matrix to be orthogonal or unitary, in order to control its eigenvalues and prevent the vanishing gradient problem. However, these had other limitations; we believe that these stem from the fact that orthogonal/unitary RNNs are also LTI. For example, they are almost always evaluated on the Copying task which they can solve perfectly, but observed to struggle on the Selective Copying task (Jing et al., 2019).

### B.4 Linear Attention

The Linear Attention (LA) (Katharopoulos et al., 2020) framework is an important result popularizing kernel attention and showing how it relates to recurrent autoregressive models. Many variants have proposed alternative kernels and other modifications. Random Feature Attention (RFA) (Peng et al., 2021) chooses the kernel feature map to approximate softmax attention (i.e. the exp feature map) using the random Fourier feature approximation of Gaussian kernels (Rahimi & Recht, 2007). Performer (Choromanski et al., 2021) finds an approximation to the exponential kernel involving only positive features, which also allows the softmax normalization term. TransNormer (Qin et al., 2022a) showed that the LA denominator term can be unstable and proposed replacing it with a LayerNorm. cosFormer (Qin et al., 2022b) augments RFA with a cosine reweighting mechanism that incorporates positional information to emphasize locality. Linear Randomized Attention (Zheng et al., 2022) generalize RFA from the perspective of importance sampling, and generalize it to provide better estimates of the full softmax kernel (rather than just the exp-transformed numerator).

Aside from kernel attention, many other variants of efficient attention exist; the survey Tay et al. (2022) offers an extensive categorization of many of these.

### B.5 Long Context Models

Long context has become a popular subject, and several recent models have claimed to scale to longer and longer sequences. However, these are often from a computational standpoint and have not been extensively validated. These include:

- Recurrent Memory Transformer (Bulatov et al., 2023), a lightweight wrapper around a Transformer backbone. It showed ability to generalize up to 1M sequences but only on synthetic memorization tasks; their main result is similar to our Induction Heads extrapolation experiment (Figure 3).
- LongNet (Ding et al., 2023), which claimed to scale to 1B length but only evaluated on length $< 100K$ for actual tasks.
- Hyena and HyenaDNA (Poli et al., 2023; Nguyen et al., 2023), which claimed to leverage up to 1M context. However, their experiments trained on proportionally more data at longer contexts, making it hard to conclude if quality improvements at 1M context are due to context length or due to more data and computation.
- Sparse Transformer (Child et al., 2019) showed a proof-of-concept of using a strided sparse attention Transformer to model audio waveforms of length $2^{20} = 1048576$, although did not discuss performance tradeoffs when controlling for computation and model size.

In contrast, we believe this work presents one of the first approaches to meaningfully demonstrate increasing performance with longer context.

## C   Mechanics of Selective SSMs

*Proof of Theorem 1.* Consider a selective SSM (Algorithm 2) with $N = 1, A = -1, B = 1, s_\Delta = $ Linear$(x), \tau_\Delta = $ softplus. The corresponding continuous-time SSM (1) is

$$h(t) = -h(t) + x(t)$$

which is also called a *leaky integrator*.

The discretization step size is

$$\Delta_t = \tau_\Delta(\text{Parameter} + s_\Delta(x_t))$$
$$= \text{softplus}(\text{Parameter} + \text{Linear}(x_t))$$
$$= \text{softplus}(\text{Linear}(x_t))$$

where we observe that the parameter can be viewed as a learnable bias and folded into the linear projection.

Now applying the zero-order hold (ZOH) discretization formulas:

$$\overline{A}_t = \exp(\Delta A) = \frac{1}{1 + \exp(\text{Linear}(x_t))} = \sigma(-\text{Linear}(x_t))$$
$$= 1 - \sigma(\text{Linear}(x_t))$$
$$\overline{B}_t = (\Delta A)^{-1}(\exp(\Delta A) - I) \cdot \Delta B = -(\exp(\Delta A) - I) = 1 - \overline{A}$$
$$= \sigma(\text{Linear}(x_t)).$$

Thus the final discrete recurrence (2a) is

$$g_t = \sigma(\text{Linear}(x_t))$$
$$h_t = (1 - g_t)h_{t-1} + g_t x_t$$

as desired.                                                                                  □

## D   Hardware-aware Algorithm For Selective SSMs

Without input-dependent selectivity, SSMs can be efficiently implemented as a convolution (Gu et al., 2022a; Dao et al., 2023), which leverages the fast Fourier transform (FFT) as primitive. With selectivity, SSMs are no-longer equivalent to convolution, but we leverage the parallel associative scan. While SSM scans are theoretically efficient ($O(BLDN)$ FLOPs, scaling linear in $L$), training foundation models with selective SSMs requires them to be efficient on modern hardware (GPUs) as well. We describe how we use *kernel fusion* and *recomputation* to make SSM scan fast and memory-efficient. We evaluate the speed of our scan implementation compared to convolution and attention in Appendix E.4, showing that it is up to $7\times$ times faster than attention at sequence length 32K, and is as memory-efficient as the best attention implementation (FlashAttention).

**Speed.** On modern hardware accelerators (GPUs) most operations (except matrix multiply) are bounded by memory-bandwidth (Williams et al., 2009; Ivanov et al., 2021; Dao et al., 2022). This the case with our scan operation, and we use kernel fusion to reduce the amount of memory IOs, leading to significant speedup compared to a standard implementation.

The standard way to implement the scan algorithm in Section 3.2 is to prepare the scan input $\overline{A},\overline{B}$ of size $(B,L,D,N)$ in GPU HBM (high-bandwidth memory, commonly referred to as GPU memory), call a parallel associative scan implementation to write the scan output of size $(B,L,D,N)$ to GPU HBM, then multiply that scan output with $C$ to produce an output of size $(B,L,D)$. However, this requires the number of memory reads/writes on the order of $O(BLDN)$. We can instead fuse the discretization step, the scan, and the multiplication with $C$ into one kernel:

1. We read in $O(BLD+DN)$ bytes of memory ($\Delta,A,B,C$) from slow HBM to fast SRAM.

2. We discretize to produce $\overline{A},\overline{B}$ of size $(B,L,D,N)$ in SRAM.

3. We perform a parallel associative scan, yielding intermediate states of size $(B,L,D,N)$ in SRAM.

4. We multiply and sum with $C$, producing outputs of size $(B,L,D)$ and write it to HBM.

This way, we reduce IOs by a factor of $O(N)$ (the state dimension), which in practice speeds up the operation by 20-40 times (Appendix E.4).

For sequence length $L$ too long where we cannot fit the sequence in SRAM (which is much smaller than HBM), we split the sequences into chunks and perform the fused scan on each chunk. As long as we have the intermediate scan states, we can continue the scan with the next chunk.

**Memory.** We describe how we use the classical technique of *recomputation* to reduce the total amount of memory required to train selective SSM layers.

From the way we fuse the forward pass, we do not save the intermediate states of size $(B,L,D,N)$ to avoid memory blowup. However, these intermediate states are necessary for the backward pass to compute gradients. We instead recompute those intermediate states in the backward pass. Since the inputs $\Delta,A,B,C$ and output gradient read from HBM to SRAM are of size $O(BLN+DN)$, and the input gradients are also of size $O(BLN+DN)$, recomputation avoids the cost of reading $O(BLND)$ elements from HBM. This means that recomputation of the SSM states in the backward pass speeds up the computation compared to storing them and reading them from HBM.

Beyond optimizing for the memory requirement of just the scan operation, we also use recomputation to optimize the memory requirement of the entire selective SSM block (input projection, convolution, activation, scan, output projection). In particular, we do not save intermediate activations that take a lot of memory but are fast to recompute (e.g. output of activation function or short convolution). As a result, the selective SSM layer has the same memory requirement as an optimized Transformer implementation with FlashAttention. In particular, each attention layer (FlashAttention) stores around 12 bytes of activations per token, an each MLP layer stores around 20 bytes of activations per token, for a total of 32 bytes ((assuming mixed-precision training in FP16 or BF16)). Each selective SSM stores around 16 bytes of activations per token. Hence two layers of selective SSMs have around the same activation memory as an attention layer and an MLP layer.

## E Full Experiments

### E.1 Synthetic Tasks

Full experiment details for these tasks including task details and training protocol are in Appendix F.1.

### E.1.1 Selective Copying

The Copying task is one of the most well-studied synthetic tasks for sequence modeling, originally designed to test the memorization abilities of recurrent models. As discussed in Section 3.1, LTI SSMs (linear recurrences and global convolutions) can easily solve this task by only keeping track of time instead of reasoning about the data; for example, by constructing a convolution kernel of exactly the right length (Figure 1). This was explicitly validated in earlier work on global convolutions (Romero et al., 2021). The Selective Copying task prevents this shortcut by randomizing the spacing between tokens. Note that this task has been introduced before as the Denoising task (Jing et al., 2019).

Note that many previous works argue that adding architecture gating (multiplicative interactions) can endow models with "data-dependence" and solve related tasks (Dao et al., 2023; Poli et al., 2023). However, we find this explanation insufficient intuitively because such gating does not interact along the sequence axis, and cannot affect the spacing between tokens. In particular architecture gating is not an instance of a selection mechanism (Appendix A).

Table 1 confirms that gated architectures such as H3 and Mamba only partially improve performance, while the selection mechanism (modifying S4 to S6) easily solves this task, particularly when combined with these more powerful architectures.

### E.1.2 Induction Heads

Induction heads (Olsson et al., 2022) is a simple task from the mechanistic interpretability lens (Elhage et al., 2021) that is surprisingly predictive of the in-context learning ability of LLMs. It requires models to perform associative recall and copy: for example, if the model has seen a bigram such as "Harry Potter" in the sequence, then the next time "Harry" appears in the same sequence, the model should be able to predict "Potter" by copying from history.

**Dataset.** We train a 2-layer model on the induction heads task at sequence length 256, with a vocab size of 16, which is comparable to prior work on this task (Dao et al., 2023) but with longer sequences. We additionally investigate generalization and extrapolation abilities by evaluating on a range of sequence lengths from $2^6 = 64$ up to $2^{20} = 1048576$ at test time.

**Models.** Following established work on induction heads, we use 2 layer models, which allows attention to mechanistically solve the induction heads task (Olsson et al., 2022). We test both multi-head attention (8 heads, with various positional encodings) and SSM variants. We use a model dimension $D$ of 64 for Mamba and 128 for the other models.

**Results.** Figure 3 shows that Mamba—or more precisely, its selective SSM layer—has the ability to solve the task perfectly because of its ability to selectively remember the relevant token while ignoring everything else in between. **It generalizes perfectly to million-length sequences, or $4000\times$ longer than it saw during training**, while no other method goes beyond $2\times$.

Out of positional encoding variants for attention models, xPos (which was designed for length extrapolation) is slightly better than the others; also note that all attention models were only tested up to sequence length $2^{14} = 16384$ due to memory limitations. Out of other SSMs, H3 and Hyena are similar, contrary to the findings in Poli et al. (2023).

### E.2 DNA Modeling

For pretraining, we largely follow a standard causal language modeling (next token prediction) setup for the training and model details (see also Appendix F.2). For the dataset, we largely follow the setup of HyenaDNA (Nguyen et al., 2023), which uses the HG38 dataset for pretraining consisting of a single human genome with about 4.5 billion tokens (DNA base pairs) in the training split.

### E.2.1 Scaling: Model Size

In this experiment, we investigate the scaling properties of genomics foundation models with various model backbones (Figure 5 *Left*).

**Training.** To advantage the baselines, we train on a short sequence length of 1024; as shown in Appendix E.2.2, we expect results to favor Mamba even more at longer sequence lengths. We fix a global batch size of 1024, for a total of $2^{20} \approx 1M$ tokens per batch. Models were trained for $10K$ gradient steps for a total of $10B$ tokens.

**Results.** Figure 5 (*Left*) shows that Mamba's pretraining perplexity improves smoothly with model size, and that Mamba scales better than both HyenaDNA and Transformer++. For example, at the largest model size of $\approx 40M$ parameters, the curve shows that **Mamba can match the Transformer++ and HyenaDNA models with roughly $3\times$ to $4\times$ fewer parameters**.

### E.2.2 Scaling: Context Length

In the next DNA experiment, we investigate the scaling properties of models with respect to sequence length. We only compare the HyenaDNA and Mamba models, as quadratic attention

becomes prohibitively expensive at longer sequence lengths. We pretrain models on sequence lengths $2^{10} = 1024$, $2^{12} = 4096$, $2^{14} = 16384$, $2^{16} = 65536$, $2^{18} = 262144$, $2^{20} = 1048576$. We fix a model size of 6 layers by width 128 (about 1.3M-1.4M parameters). Models were trained for $20K$ gradient steps for a total of $\approx 330B$ tokens. The longer sequence lengths used sequence length warmup similar to (Nguyen et al., 2023).

**Results.** Figure 5 (*Right*) shows that **Mamba is able to make use of longer context even up to extremely long sequences of length 1M**, and its pretraining perplexity improves as the context increases. On the other hand, the HyenaDNA model gets worse with sequence length. This is intuitive from the discussion in Section 3.5 on properties of the selection mechanism. In particular, LTI models cannot selectively ignore information; from a convolutional perspective, a very long convolution kernel is aggregating all information across a long sequence which may be very noisy. Note that while HyenaDNA claims to improve with longer context, their results do not control for computation time.

### E.2.3 Synthetic Species Classification

We evaluate models on a downstream task of classifying between 5 different species by randomly sampling a contiguous segment of their DNA. This task is adapted from HyenaDNA, which used the species {human,lemur,mouse,pig,hippo}. We modify the task to be significantly more challenging by classifying between the five *great apes* species {human,chimpanzee,gorilla,orangutan,bonobo}, which are known to share 99% of their DNA.

### E.3 Model Ablations

We perform a series of detailed ablations on components of our model, focusing on the setting of language modeling with size $\approx 350M$ models at Chinchilla token counts (same setting as Figure 4).

### E.3.1 Architecture

Table 5 investigates the effects of the architecture (block) and its inner SSM layer (Figure 2). We find that

- Among previous non-selective (LTI) SSMs, which are equivalent to global convolutions, performance is very similar.
- Replacing the complex-valued S4 variant from previous work with a real-valued one does not affect performance much, suggesting that (at least for LM) real-valued SSMs may be a better choice when accounting for hardware efficiency.
- Replacing any of these with a selective SSM (S6) significantly improves performance, validating the motivation of Section 3.
- The Mamba architecture performs similarly to the H3 architecture (and seems slightly better when using a selective layer).

We also investigate interleaving the Mamba block with other blocks such as MLP (a traditional architecture) MHA (a hybrid attention architecture) in Appendix F.2.2.

### E.3.2 Selective SSM

Table 6 ablates the selective SSM layer by considering different combinations of selective $\Delta$, $B$, and $C$ parameters (Algorithm 2), showing that $\Delta$ is the most important parameter due to its connection to RNN gating (Theorem 1).

Table 7 considers different initializations of the SSM, which have been shown to make a large difference in some data modalities and settings (Gu et al., 2022a;b). On language modeling, we find that simpler real-valued diagonal initializations (S4D-Real, row 3) instead of more standard complex-valued parameterizations (S4D-Lin, row 1) perform better. Random initializations also work well, consistent with findings from prior work (Mehta et al., 2023).

Table 8 and Table 9 consider varying the dimension of the $\Delta$ and ($B,C$) projections respectively. Changing them from static to selective provides the most benefit, while increasing the dimensions further generally improves performance modestly with a small increase in parameter count.

Of particular note is the dramatic improvement of the selective SSM when the state size $N$ is increased, with over a 1.0 perplexity improvement for a cost of only 1% additional parameters. This validates our core motivation in Sections 3.1 and 3.3.

Table 5: (**Ablations: Architecture and SSM layer**.) The Mamba block performs similarly to H3 while being simpler. In the inner layer, there is little difference among different parameterizations of LTI models, while selective SSMs (S6) provide a large improvement. More specifically, the S4 (real) variant is S4D-Real and the S4 (complex) variant is S4D-Lin.

| MODEL | ARCH. | SSM LAYER | PPL | MODEL | ARCH. | SSM LAYER | PPL |
|-------|-------|-----------|-----|-------|-------|-----------|-----|
| Hyena | H3 | Hyena | 10.24 | - | Mamba | Hyena | 10.75 |
| H3 | H3 | S4 (complex) | 10.30 | - | Mamba | S4 (complex) | 10.54 |
| - | H3 | S4 (real) | 10.34 | - | Mamba | S4 (real) | 10.56 |
| - | H3 | S6 | **8.95** | Mamba | Mamba | S6 | **8.69** |

Table 6: (**Ablations: Selective parameters**.) $\Delta$ is the most important parameter (Theorem 1), but using multiple selective parameters together synergizes.

| SELECTIVE $\Delta$ | SELECTIVE $B$ | SELECTIVE $C$ | PPL |
|-------------------|--------------|--------------|-----|
| ✗ | ✗ | ✗ | 10.93 |
| ✗ | ✓ | ✗ | 10.15 |
| ✗ | ✗ | ✓ | 9.98 |
| ✓ | ✗ | ✗ | 9.81 |
| ✓ | ✓ | ✓ | 8.71 |

Table 7: (**Ablations: Parameterization of** $A$.) The more standard initializations based on S4D-Lin (Gu et al., 2022b) perform worse than S4D-Real or a random initialization, when the SSM is selective.

| $A_n$ INITIALIZATION | FIELD | PPL |
|----------------------|-------|-----|
| $A_n = -\frac{1}{2} + ni$ | Complex | 9.16 |
| $A_n = -1/2$ | Real | 8.85 |
| $A_n = -(n+1)$ | Real | 8.71 |
| $A_n \sim \exp(\mathcal{N}(0,1))$ | Real | 8.71 |

Table 8: (**Ablations: Expressivity of** $\Delta$.) The selection mechanism of $\Delta$ constructs it with a projection of the input. Projecting it even to dim. 1 provides a large increase in performance; increasing it further provides further improvements at the cost of a modest increase in parameters. State size fixed to $N = 16$.

| SIZE OF $\Delta$ PROJ. | PARAMS (M) | PPL |
|------------------------|------------|-----|
| - | 358.9 | 9.12 |
| 1 | 359.1 | 8.97 |
| 2 | 359.3 | 8.97 |
| 4 | 359.7 | 8.91 |
| 8 | 360.5 | 8.83 |
| 16 | 362.1 | 8.84 |
| 32 | 365.2 | 8.80 |
| 64 | 371.5 | 8.71 |

Table 9: (**Ablations: SSM state dimension**.) (*Top*) Constant $B$ and $C$ (*Bottom*) Selective $B$ and $C$. Increasing the SSM state dimension $N$, which can be viewed as an expansion factor on the dimension of the recurrent state, can significantly improve performance for a negligible cost in parameters/FLOPs, but only when $B$ and $C$ are also selective. Size of $\Delta$ projection fixed to 64.

| STATE DIMENSION $N$ | PARAMS (M) | PPL |
|---------------------|------------|-----|
| 1 | 367.1 | 9.88 |
| 2 | 367.4 | 9.86 |
| 4 | 368.0 | 9.82 |
| 8 | 369.1 | 9.82 |
| 16 | 371.5 | 9.81 |
| 1 | 367.1 | 9.73 |
| 2 | 367.4 | 9.40 |
| 4 | 368.0 | 9.09 |
| 8 | 369.1 | 8.84 |
| 16 | 371.5 | 8.71 |

### E.4  Speed and Memory Benchmarks

We benchmark the speed of the SSM scan operation (state expansion $N = 16$), as well as the end-to-end inference throughput of Mamba, in Figure 8. Our efficient SSM scan is faster than the best attention implementation that we know of (FlashAttention-2 (Dao, 2024)) beyond sequence length 2K, and up to 20-40× faster than a standard scan implementation in PyTorch. Mamba achieves 4-5× higher inference throughput than a Transformer of similar size, since without the KV cache it can use much higher batch sizes. For example, a Mamba-6.9B (untrained) would have higher inference throughput than a 5× smaller Transformer-1.3B. Details in Appendix F.5, which additionally includes a benchmark of memory consumption.

Table 10: (**Induction heads**.) Models are trained on sequence length $2^8 = 256$, and tested on various sequence lengths of $2^6 = 64$ up to $2^{20} = 1048576$. ✓ denotes perfect generalization accuracy, while ✗ denotes out of memory.

| MODEL | PARAMS | TEST ACCURACY (%) AT SEQUENCE LENGTH | | | | | | | | | | | | | | |
|---|---|---|---|---|---|---|---|---|---|---|---|---|---|---|---|---|
| | | $2^6$ | $2^7$ | $\mathbf{2^8}$ | $2^9$ | $2^{10}$ | $2^{11}$ | $2^{12}$ | $2^{13}$ | $2^{14}$ | $2^{15}$ | $2^{16}$ | $2^{17}$ | $2^{18}$ | $2^{19}$ | $2^{20}$ |
| MHA-Abs | 137K | ✓ | 99.6 | 100.0 | 58.6 | 26.6 | 18.8 | 9.8 | 10.9 | 7.8 | ✗ | ✗ | ✗ | ✗ | ✗ | ✗ |
| MHA-RoPE | 137K | ✓ | ✓ | 100.0 | 83.6 | 31.3 | 18.4 | 8.6 | 9.0 | 5.5 | ✗ | ✗ | ✗ | ✗ | ✗ | ✗ |
| MHA-xPos | 137K | ✓ | ✓ | 100.0 | 99.6 | 67.6 | 25.4 | 7.0 | 9.0 | 7.8 | ✗ | ✗ | ✗ | ✗ | ✗ | ✗ |
| H3 | 153K | ✓ | ✓ | 100.0 | 80.9 | 39.5 | 23.8 | 14.8 | 8.2 | 5.9 | 6.6 | 8.2 | 4.7 | 8.2 | 6.3 | 7.4 |
| Hyena | 69M* | 97.7 | ✓ | 100.0 | ✓ | 44.1 | 12.5 | 6.6 | 5.1 | 7.0 | 5.9 | 6.6 | 6.6 | 5.9 | 6.3 | 9.8 |
| Mamba | 74K | ✓ | ✓ | 100.0 | ✓ | ✓ | ✓ | ✓ | ✓ | ✓ | ✓ | ✓ | ✓ | ✓ | ✓ | ✓ |

\* Most of the parameters are in learnable positional encodings.

Table 11: (**Scaling Law Model Sizes**.) Our model sizes and hyperparameters for scaling experiments. (Model dimension and number of heads applies only to Transformer models.)

| PARAMS | n_layers | d_model | n_heads / d_head | TRAINING STEPS | LEARNING RATE | BATCH SIZE | TOKENS |
|---|---|---|---|---|---|---|---|
| 125M | 12 | 768 | 12 / 64 | 4800 | 6e-4 | 0.5M tokens | 2.5B |
| 350M | 24 | 1024 | 16 / 64 | 13500 | 3e-4 | 0.5M tokens | 7B |
| 760M | 24 | 1536 | 16 / 96 | 29000 | 2.5e-4 | 0.5M tokens | 15B |
| 1.3B | 24 | 2048 | 32 / 64 | 50000 | 2e-4 | 0.5M tokens | 26B |

# F  Experimental Details and Additional Results

## F.1  Synthetic Tasks

**Selective Copying.**  Our setting is on sequences of length 4096, with a vocab size of 16 possible tokens (including the white "noise" token from Figure 1) and requiring models to memorize 16 "data" tokens. We use 2 layer models with a model dimension of $D = 64$.

Models are trained for 400K steps at a constant learning rate of 0.0001 with a batch size of 64.

**Induction Heads.**  Training consists of randomly generating data every step, with a batch size of 8. We choose an "epoch" size of 8192 steps, and track the accuracy on fixed validation sets (also randomly generated) of each target sequence length. For the MHA-Abs and Mamba models, results are reported after the 25th epoch ($8192 \times 25 = 204800$ steps). For the MHA-RoPE and MHA-xPos models, results are reported after the 50th epoch ($8192 \times 50 = 409600$ steps). For the LTI H3 and Hyena models, results are reported after the 10th epoch (81920 steps) because they had converged by then and failed to improve further.

We use the Adam optimizer with no weight decay. All models are trained at constant learning rates $2e-4$ and $1e-3$, and the better results are reported for each model ($2e-4$ for all models except Mamba). The attention and Hyena models did not learn at LR $1e-3$. H3 learned at both LRs, but interestingly generalized better to shorter sequences at the smaller LR of $2e-4$. Mamba learned at both LRs, but extrapolated better at the larger LR of $1e-3$.

## F.2  Language Modeling

### F.2.1  Scaling Law Details

Scaling law experiments generally followed the GPT3 recipe. All models were trained on the Pile with the GPT2 tokenizer.

**Model Sizes.**  Table 11 specifies the model sizes we use for scaling laws. This is taken directly from the GPT3 specifications (Brown et al., 2020), with very minor modifications. First, we changed the batch size of the 1.3B model from 1M tokens to 0.5M tokens, since we did not use enough parallelization to require the larger batch size. Second, we changed the number of training steps and total tokens to roughly match Chinchilla scaling laws (Hoffmann et al., 2022), which specify that training tokens should increase proportionally to model size.

**Training Recipes.**  All models used the AdamW optimizer with

- gradient clip value 1.0

- weight decay 0.1

- no dropout

- linear learning rate warmup with cosine decay

By default, the peak learning rate is the GPT3 specification.

We give several models an "improved recipe", inspired by changes adopted by popular large language models such as PaLM (Chowdhery et al., 2023) and LLaMa (Touvron et al., 2023). These include:

- linear learning rate warmup with cosine decay to $1e-5$, with a peak value of $5\times$ the GPT3 value

- no linear bias terms

- RMSNorm instead of LayerNorm

- AdamW hyperparameter $\beta=(.9,.95)$ (the GPT3 value) instead of the PyTorch default of $\beta=(.9,.999)$

**Architecture and Training Details.**    Our models are:

- **Transformer**: The standard Transformer based on GPT3 (Table 11).
- **Transformer++**: A Transformer with an improved architecture, namely rotary positional encodings (Su et al., 2021) and SwiGLU MLP (Shazeer, 2020), and the improved training recipe above.
- **Hyena**: Interleaving a Hyena block (the H3 block with S4 replaced by a global convolution parameterized by an MLP) with standard MLP blocks. The MLP blocks have expansion factor 2 instead of 4 and the number of layers is correspondingly increased by $1.5\times$ to preserve parameter count.
- **H3++**: The H3 architecture with a few modifications, including (i) using the same "thin" Hyena dimensions above (ii) the improved training recipe above (iii) a linear attention *head dimension* of 8.
- **RWKV**: The default RWKV model from Peng et al. (2023), including its modified MLP block. We also used as much of its specified training recipe as possible, such as increasing the learning rates by $2\times$ or $3\times$ on certain parameters.
- **RetNet**: The default RetNet model from Sun et al. (2023). We also gave it the improved training recipe above.
- **Mamba**: The standard Mamba architecture, with the improved training recipe.

F.2.2    Additional Scaling Law Ablations

We perform additional ablations on the architecture using the same protocol as the 2k context length scaling laws in Figure 4 (*Left*).

**Mamba Architecture: Interleaving Blocks.**    We test the effect of different architectural blocks combined with the Mamba block. We focus on the viewpoint that the Mamba block is simply the standard SwiGLU block with an extra conv $\rightarrow$ SSM path added. This leads to two natural ablations:

- What if the Mamba block is interleaved with a standard MLP block, instead of stacked homogenously? This can also be interpreted as taking Mamba and removing half of the SSMs.
- What if the Mamba block is interleaved with MHA (multi-head attention) blocks? This can also be interpreted as taking a Transformer with SwiGLU MLPs (i.e. what we call Transformer++) and simply adding SSMs to the MLP blocks.

Figure 9 (*Right*) shows these variants compared to the original (homogenous) Mamba architecture. Interestingly, neither change matters too much. The Mamba-MLP architecture is only slightly worse, and still better than all models except Transformer++. The Mamba-MHA architecture is only slightly better, which is somewhat surprising in light of the fact that many recent works have found that combining (LTI) SSMs with Attention can lead to substantial improvements (Dao et al., 2023; Fathullah et al., 2023; Saon et al., 2023; Zuo et al., 2022; Fathi et al., 2023).

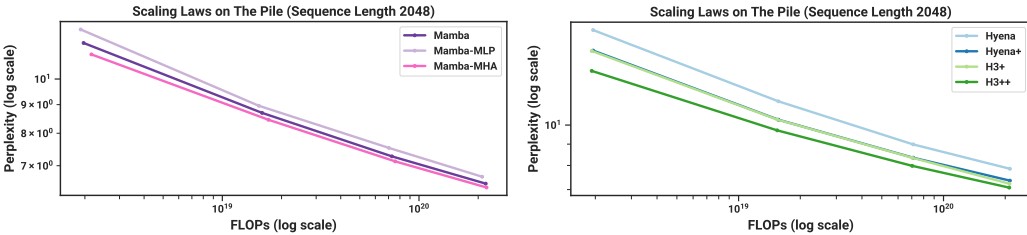

Figure 9: (**Scaling laws: extra ablations**.) (*Left*) Instead of (*Right*) Instead of

**H3 Architecture: Training Recipes.** Next we ablate differences between the Hyena and H3++ models, our weakest and strongest models outside of Transformer++ and Mamba, particularly to isolate the effect of training recipes.

- **Hyena**: The Hyena block with its original architecture and GPT3 training recipe (same as Figure 4).
- **Hyena+**: The same architecture but with the improved training recipe described above.
- **H3+**: The same architecture as Hyena+ but with the Hyena convolution kernel swapped out for S4D convolution kernel.
- **H3++**: The same as H3+, but with a linear attention *head dimension* of 8. This increases computation inside the SSM recurrence but does not increase parameters.

Our general convention is that "Model+" represents the base model with the improved training recipe, and "Model++" also allows for architectural changes.

Figure 9 (*Right*) shows that

- A large improvement is achieved by the improved training recipe, which was used for many of the models in the main Figure 4 (RetNet, H3++, Transformer++, Mamba).
- The choice of the inner LTI SSM does not matter (e.g. Hyena vs. S4), consistent with findings throughout this paper.
- The head dimension expansion improves performance, consistent with one of our main themes that expanded state dimension improves performance for SSMs (Section 3).

### F.2.3 Downstream Evaluation Details

This pretraining procedure is the same as the scaling law protocol, but extended to 300B tokens and with the GPT-NeoX tokenizer (Black et al., 2022) instead of GPT2 tokenizer. For the 1.3B model, we use a batch size of 1M tokens to be consistent with the GPT3 specifications. We report the perplexity on the Pile validation set, and for this metric only compare to models trained on the same dataset and with the same tokenizer, in particular Pythia and RWKV.

For downstream evaluation, we use the LM evaluation harness from EleutherAI (Gao et al., 2021), as done by most work in this area. We evaluate on the following tasks/datasets that measure common sense reasoning:

- LAMBADA (Paperno et al., 2016)
- HellaSwag (Zellers et al., 2019)
- PIQA (Bisk et al., 2020)
- ARC-challenge (Clark et al., 2018)
- ARC-easy: an easy subset of ARC-challenge
- WinoGrande (Sakaguchi et al., 2021)

We report accuracy for LAMBADA, WinoGrande, PIQA, and ARC-easy, and accuracy normalized by sequence length for HellaSwag and ARC-challenge (since normalized accuracy is higher for almost all models for these task).

### F.3 DNA Modeling

### F.3.1 Pretraining Details

We describe the dataset and training procedure of the HG38 pretraining task in more detail.

The dataset follows the splits from the prior Enformer work on genomics (Avsec et al., 2021); the training split contains a total of $S = 34021$ segments of length $2^{17} = 131072$ that cover the genome, for a total of approximately 4.5 billion tokens (DNA base pairs). These segments are pairs of (chromosome number, starting index, ending index), and can be extended if necessary (e.g. to get longer segments).

We deviate from HyenaDNA when the training sequence length is not $2^{17}$. HyenaDNA always takes a fixed sub-segment (e.g. the beginning or middle of the prescribed segment), and thus for any training sequence length each epoch is fixed to 34021 samples and doesn't necessarily go through the whole genome. On the other hand, we use the entire training data:

- When the context length $L$ is less than (or equal to) $2^{17}$, we divide up each segment into non-overlapping sub-segments of length $L$, so that there are $S \times \frac{2^{17}}{L}$ total samples and $S \times 2^{17} \approx 4.5B$ tokens per epoch.

- When the context length $L$ is greater than $2^{17}$, we turn each segment into two samples, one that begins with the prescribed segment and one that ends with the prescribed segment. Thus each epoch has $2S$ items and $2SL$ tokens per epoch. For example, at sequence length $2^{18} = 262144$ there are $4\times$ as many tokens as the default, and at sequence length $2^{20}$ there are $16\times$ as many tokens.

Other training details generally follow the same protocol as our language modeling experiments (Appendix F.2). For example, we use the AdamW with $(\beta_1, \beta_2) = (0.9, 0.95)$, no dropout, weight decay 0.1. We use a cosine learning rate scheduler with linear warmup for 10% of total steps.

### F.3.2 Scaling: Model Size Details

**Models.** The models we consider are:

- Transformer++: a Transformer with improved architecture, notably the usage of RoPE positional encodings (Su et al., 2021). Informally, we found these to be noticeably better than vanilla positional encodings from (Vaswani et al., 2017).
- HyenaDNA: the Hyena model from Poli et al. (2023); Nguyen et al. (2023), which is roughly a Transformer with the MHA block replaced by an H3 block using a global convolution parameterized by an MLP.
- Mamba: the standard Mamba architecture.

**Model Sizes.** We use the following model sizes.

| BLOCKS | 4 | 5 | 6 | 7 | 8 | 10 | 12 |
|---|---|---|---|---|---|---|---|
| MODEL DIMENSION | 64 | 96 | 128 | 192 | 256 | 384 | 512 |
| PARAMS (APPROX.) | 250K | 700K | 1.4M | 3.5M | 7.0M | 19.3M | 40.7M |

Note that the number of blocks for Mamba is doubled, because one Transformer "layer" includes both the MHA and MLP blocks (and similarly for Hyena), which requires two Mamba blocks to match parameters (Section 3.4).

**Training.** For each model (Transformer++, HyenaDNA, Mamba), we swept the learning rate across $\{1e-3, 2e-3, 4e-3, 8e-3\}$. The optimal Transformer and HyenaDNA learning rates were 2e-3 across all sizes. The optimal Mamba learning rate was 8e-3; note that Mamba performed better than baselines with matched learning rates (2e-3), but was more stable and improved even more at higher learning rates. (Furthermore, as this LR is on the upper range of the sweep, it is possible that our results are still suboptimal.)

Note that, in contrast to standard LM scaling laws (Table 11), our LR held constant across model sizes for simplicity. The optimal LR should go down for larger models, but we didn't find a noticeable effect at the small model sizes (at most a few million parameters) we considered.

### F.3.3  Scaling: Context Length Details

We use a total batch size of $2^{24} \approx 16M$ tokens per training step, for every sequence length (e.g. at length $2^{20}$ there are 16 segments per batch and at length $2^{10}$ there are 16384 segments per batch). This is a large batch size relative to the model size by usual LM standards, but note that a batch size of $2^{23}$ is the minimum possible on a machine with 8 GPUs and sequence length of $2^2 0$, and that HyenaDNA used much larger batches of $2^{28}$.

The learning rate used was 0.008 for Mamba and 0.001 for HyenaDNA; we initially attempted to use the same learning rate of 0.002 from the previous section for HyenaDNA, but found that it was unstable at the longest context length.

**Sequence Length Warmup.**  Following (Nguyen et al., 2023), we use sequence length warmup (SLW) during pretraining. We choose a simple schedule of 2 epochs at each power-of-two sequence length starting from $2^{10} = 1024$. (Note that because of how data is curated, at the longest sequence lengths more steps and tokens are spent proportionally. In particular, each stage up to length $2^{17}$ processes the same number of tokens, but $4\times$ as many tokens are processed at length $2^{18}$, $8\times$ as many at length $2^{19}$, and $16\times$ as many at length $2^{20}$.)

Unlike HyenaDNA, we always control for the number of tokens per gradient update, so the batch size is successively halved as the sequence lengths are doubled in each stage.

**Remark F.1.**  *We also note that the schedule was not tuned, and we never experimented with turning off sequence length warmup for these pretraining experiments. We later found that SLW did not help noticeably for audio pretraining at similar lengths (Section 4.4), and it is possible that it is not necessary for DNA pretraining either.*

### F.3.4  Species (Great Apes) Classification

Models are causal and therefore only the last element (across the sequence length) of the model's output is used for the classification head. Note that we control for the total number of elements in the loss function per gradient step. The pretraining objective includes all positions across the sequence length, so that `batch_size` $\times$ `sequence_length` is held constant; in other words, the batch size decreases as the sequence length increases. However, for a classification task, since only the last position enters the loss, the batch size itself is held constant. Note that this also means that fine-tuning models with longer sequence lengths is more computationally expensive.

Training consists of 10 epochs, each of which has 1024 gradient steps. Each gradient step uses batch size 64, which are all independently randomly drawn by uniformly picking a species, uniformly picking a chromosome, and then uniformly picking a contiguous segment of DNA.

Following (Nguyen et al., 2023), models with a maximum context length greater than $2^{14} = 16384$ use sequence length warmup with 1 epoch at length $2^{14} = 16384$, 1 epoch at length $2^{15} = 32768$, 1 epoch at length $2^{16} = 65536$, and so on up to the maximum sequence length. For example, the model with $2^{20} = 1048576$ context undergoes 6 epochs of sequence length warmup before 4 more epochs at its maximum sequence length.

The learning rate for all Hyena models is $4e-5$, while the learning rate for all Mamba models is $1e-4$. These were found by performing learning rate sweeps for each model among $\{1e-5, 2e-5, 4e-5, 1e-4, 2e-4\}$ for the smaller sequence lengths ($2^{10}, 2^{12}, 2^{14}, 2^{16}$), and these values were consistently found to be the best for each model. An abridged learning rate sweep was done at length $2^{18}$, which agreed with these values, and a single run at length $2^{20}$ was performed (as described above, the computational cost of these experiments is proportional to the sequence length). The learning rate followed a cosine decay schedule with warmup with 5 epochs of linear warmup to the maximum learning rate, and 5 epochs of cosine decay down to $1e-6$. The unusually long learning rate warmup schedule was chosen because the sequence length warmup was also long (e.g. comprising 6 out of 10 epochs for the model with context length $2^{20}$); we did not experiment with this choice.

Results for the Species classification task are in Table 12.

Table 12: (**Great Apes DNA Classification**.) Accuracy after fine-tuning on sequences of length $2^{10} = 1024$ up to $2^{20} = 1048576$ using pretrained models of the same context length. Random guessing is 20%.

| MODEL | PARAMS | ACCURACY (%) AT SEQUENCE LENGTH | | | | | |
|---|---|---|---|---|---|---|---|
| | | $2^{10}$ | $2^{12}$ | $2^{14}$ | $2^{16}$ | $2^{18}$ | $2^{20}$ |
| HyenaDNA | 1.4M | 28.04 | 28.43 | 41.17 | 42.22 | 31.10 | 54.87 |
| Mamba | 1.4M | 31.47 | 27.50 | 27.66 | 40.72 | 42.41 | **71.67** |
| Mamba | 7M | 30.00 | 29.01 | 31.48 | 43.73 | 56.60 | **81.31** |

Table 13: YouTubeMix length scaling sequence lengths and batch sizes.

| SEQUENCE LENGTH | BATCH SIZE | TOKENS / BATCH |
|---|---|---|
| $468 \times 2048 = 958464$ | 1 | 958464 |
| $234 \times 2048 = 479232$ | 2 | 958464 |
| $117 \times 2048 = 239616$ | 4 | 958464 |
| $59 \times 2048 = 120832$ | 8 | 966656 |
| $30 \times 2048 = 61440$ | 16 | 983040 |
| $15 \times 2048 = 30720$ | 32 | 983040 |
| $8 \times 2048 = 16384$ | 64 | 1048576 |
| $4 \times 2048 = 8192$ | 128 | 1048576 |

### F.4 Audio Details

### F.4.1 YouTubeMix Audio Pretraining

**Model.** We use a model with 3 blocks per stage ($3 \times 5 = 15$ total Mamba blocks), pooling factor $p = 16$, and outer dimension $D = 64$, for about 3.5M parameters.

**Dataset.** The data is mu-law encoded at 8 bits, so the model is modeling discrete tokens with a vocab size of 256.

The dataset consists of clips of up to 1 minute long, or length 960000, which is subsampled and divided into segments of any desired sequence length. Since the architecture involves two stages of pooling by a factor of 16, and we want the resulting sequence length to be a a multiple of 8 for hardware efficiency, the longest possible sequence is $468 \times 2048 = 958464$. The rest of our sequence lengths are defined by successively halving this and rounding up to the nearest multiple of 2048.

Table 13 lists the specifications used in Figure 7. Beyond the varying batch sizes, the number of valid segments in the training set varied between different sequence lengths (e.g. the number of training steps per epoch was not constant for different points in the graph), which may have contributed to kinks in the scaling curves.

**Training.** Models were trained for $200K$ training steps with a maximum learning rate of 0.002, $20K$ (10%) warmup steps, and weight decay 0.1 (similar to our general pretraining recipe across domains).

**Additional Ablations: SSM Parameterizations.** We investigate SSM parameterizations on long-form audio waveform pretraining in the setting of Figure 7. The setting is modified slightly to use larger models (8 layers and $D = 64$ for 6M params, the SaShiMi default), shorter sequences ($2^{11} = 2048$ to $2^{18} = 262144$ instead of $2^{13}$ to $2^{20}$), lower LR (0.001 from 0.002), and shorter training cycles (100K instead of 200K steps).

Figure 10 shows that the change from S4 $\rightarrow$ S6 (i.e. the selection mechanism) is not always beneficial. On long-form audio waveforms, it in fact significantly hampers performance, which may be intuitive from the point of view that audio is uniformly sampled and very smooth, and therefore benefits from continuous linear time-invariant (LTI) methods. After ablating away the selection mechanism, note that the resulting model is the S4 layer inside the Mamba block. To disambiguate, we call this Mamba-S4 as opposed the default Mamba architecture Mamba-S6.

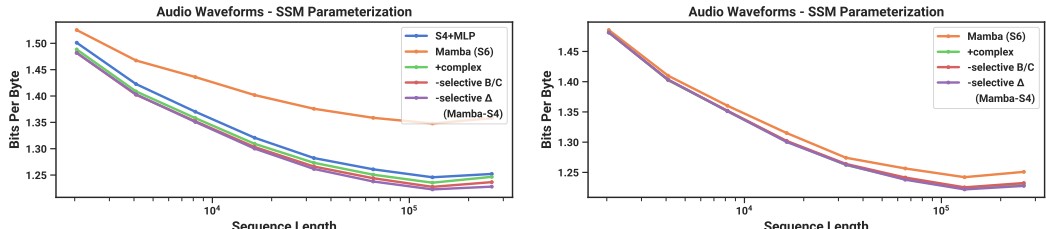

Figure 10: (**Audio Pretraining (YouTubeMix) Ablations**.) As a uniformly-sampled "continuous" signal modality, audio waveforms actually benefit from LTI models which have matching inductive bias. (*Left*) Homogenous models (all blocks have the same parameterization) (*Right*) Only the center U-Net blocks are ablated; the outer blocks are Mamba-S4. Purple line is same as figure on left.

However, on the right side, we keep the outer layers of the U-Net Mamba-S4 and ablate only the inner layers. The performance differences shrink dramatically; this reinforces the hypothesis that layers closer to the *raw* audio signal should be LTI, but once they are "tokenized" and compressed by the outer layers, the inner layers no longer need to be LTI. In this setting however, the real-valued SSM still underperforms the complex-valued one.

### F.4.2 SC09 Speech Generation

Autoregressive training largely followed the autoregressive language modeling protocol, such as

- Weight decay 0.1
- Learning rate warmup for 10% of total steps
- AdamW optimizer with $\beta = (0.9, 0.95)$
- Gradient clip value 0.1

We used a learning rate of 0.002 and 200000 training steps at a batch size of 16.

The large Mamba model in Table 3 has 15 layers per stage with an outer dimension of $D = 96$ and pooling factor 4. We note that this dataset is small (training went through 100 epochs) and for this large model, there was significant overfitting of the BPB or NLL. However, automated metrics of generated samples continually improving throughout training.

The models in the architecture ablations in Table 4 all have 8 layers per stage with an outer dimension of $D = 64$ and pooling factor 4. The S4+MLP block has roughly $2D^2 + 4D^2$ parameters (expansion factor 2 in the MLP). The Transformer block has $4D^2 + 2D^2$ parameters (expansion factor 1 in the MLP). The Mamba block has the usual $\approx 6D^2$ parameters. All models have roughly 6M total parameters.

### F.5 Efficiency Benchmark

**Scan Operation.** We compare the core operation of selective SSMs, which is the parallel scan (Section 3.3), against convolution and attention, measured on an A100 80GB PCIe GPU. Note that these do not include the cost of other operations outside of this core operation, such as computing the convolutional kernel in global-convolution models, or computing the QKV projections in attention.

As a baseline, we implement a standard parallel scan in PyTorch with no kernel fusion. This requires materializing the parameters $\overline{A}, \overline{B}, C$ in HBM.

Our scan implementation fuses the discretization step and the parallel scan, avoiding the cost of materializing all the large parameters in HBM.

For convolution, we use the standard implementation in PyTorch, which separately performs FFTs on the inputs and the filters, multiply them in frequency domain, then performs an inverse FFT to obtain the result. The theoretical complexity is $O(L \log(L))$ for sequence length $L$.

For attention, we compare against the fastest implementation that we are aware of (FlashAttention-2 (Dao, 2024)), with causal mask. Note that FlashAttention-2 with causal mask is about $1.7 \times$ faster than without causal mask, since approximately only half of the attention entries are computed.

We use batch size of 1 and increase the sequence length from $2^9 = 512$, $2^{10} \approx 1K$, $2^{11} \approx 2K$, up to $2^{19} \approx 500K$ (some of the baselines run out of memory before reaching 500K). We use a model

Table 14: (**Memory benchmark**.) Mamba's memory footprint is comparable to the most optimized Transformer. Results for 125M models.

| Batch size | Transformer (w/ FlashAttention-2) | Mamba |
|---|---|---|
| 1 | 4.6GB | 4.8GB |
| 2 | 5.2GB | 5.8GB |
| 4 | 6.9GB | 7.3GB |
| 8 | 11.5GB | 12.3GB |
| 16 | 20.7GB | 23.1GB |
| 32 | 34.5GB | 38.2GB |

dimension of $D = 1024$ and state dimension $N = 16$. We measure with BF16 inputs, which is the data type most commonly used for large scale training.

**End-to-end Inference.**    We measure the inference throughput of a Mamba 1.4B model and an untrained Mamba 6.9B model, against a standard Transformer (GPT3 architecture) at 1.3B and 6.7B size. We use the standard Transformer implementation in the Huggingface `transformers` library.

We set the prompt length to be 2048 and the generation length to be 128. We vary the batch size from 1, 2, 4, 8, 16, 32, 64, to 128, and measure time time taken to generate 128 tokens. We then calculate the throughput (tokens/s) as batch size $\times 128$/time taken. We repeat the measurements 3 times and take the average. Measurements are done on an A100 80GB PCIe GPU.

**Memory Benchmark.**    The memory usage simply scales proportionally to the size of the activation tensors, as with most deep sequence models. We report measurements of the training memory requirements of 125M models on 1 A100 80GB GPU. Each batch consists of sequences of length 2048. We compare to the most memory-efficient Transformer implementation we are aware of (with kernel fusion from `torch.compile` and with FlashAttention-2). Table 14 shows that Mamba's memory requirement is comparable to a similar-sized Transformer with an extremely optimized implementation, and we expect further improvement in Mamba's memory footprint in the future.

