# OpenReview forum: "Mamba: Linear-Time Sequence Modeling with Selective State Spaces"
_colmweb.org/COLM/2024/Conference — COLM_

### Official Review · Reviewer_pB6t · 2024-05-11

**Rating:** 10
**Confidence:** 5
**Ethics Flag:** 1

**Summary:**

Previous state-space models (SSMs) aimed to address the inefficiencies of Transformer architectures over long sequences, but they generally underperformed compared to Transformers. This paper introduces a new SSM architecture, Mamba, which has efficient long-context language modeling with linear complexity. Compared to S4, Mamba incorporates a selection mechanism that allows the model to selectively memorize information based on previous inputs. The architecture also combines Mamba layers with MLP blocks to further improve computational efficiency. The implementation uses a hardware-aware design that efficiently utilizes SRAM for discretization and recurrence.

In synthetic tasks such as selective copying and induction heads, Mamba demonstrated significant improvements over previous SSMs, including H3 and S4 models. These enhancements show Mamba's capability to handle long context effectively. Further experimentation with models scaling up to 2.8 billion parameters confirmed Mamba's robustness on language modeling, showing that it achieves comparable performance to Transformer models with the same amount of training data. The authors have conducted experiments extends beyond language modeling with experiments on audio and genomics data, which illustrates its great performance in processing long sequences across different modalities.

**Questions To Authors:**

* Compression is a key concept in SSMs and recurrent models, as history needs to be stored in fixed-size states, but it would limit the memory capacity. For Transformers, KV cache can grow infinitely, so theoretically they have unlimited memory capacity. It is interesting to see what is the upper-bound of Mamba's memory capacity in terms of memorizing history information.

*  The current implementation doesn't support passing in initial state, and getting their gradients during back-propagation. This may help to solve the length generalization problem of Mamba, and potentially can be combined with gradient check-pointing in the sequence dimension to train the model with very long sequence length.

**Reasons To Accept:**

* Mamba has solved the implementation difficulty of previous SSMs with fused kernel and hardware-aware algorithm.

* The authors have conducted extensive experiments on Mamba with different settings.

* Mamba has greatly contributed to the development of efficient LLMs and SSMs. It has the potential to become the successor of Transformers.

**Reasons To Reject:**

There is no reason to reject this paper.

---

> ### Author Rebuttal · Authors · 2024-05-31
>
> Thanks for the encouragement!
>
> > State compression and memory capacity
>
> Absolutely – we believe that the tradeoff between state size and model capacity is central to sequence models, and there are many exciting research directions leading from here.
> * The Based paper [1] found that, when plotting memorization capability against effective state size, that various finite-state models including SSMs actually have comparable performance to attention (as a function of state size). Of course, pure softmax attention can get larger effective state sizes.
> * Recent SSM follow-ups have found ways to get much larger state sizes (GLA [2], HGRN2 [3], Mamba-2 [4], RWKV-6 [5], xLSTM [6]), which significantly improves their performance on synthetic memorization tasks.
> * We think that attempting to theoretically characterize the memory capacity of models in terms of their effective state size (e.g. proving a lower bound) would be a very interesting theoretical question on sequence models.
> * Another interesting direction for improving models is whether a model’s state size can grow with the context length. For example, the linear growth (i.e. KV cache) of a pure Transformer might be wasteful, whereas the constant size state of a pure SSM might be too restrictive; can a model’s state grow slowly through the sequence length instead?
>
> [1] Simran Arora et al. “Zoology: Measuring and Improving Recall in Efficient Language Models” ICML 2024.
>
> [2] Songlin Yang et al. “Gated Linear Attention Transformers with Hardware-Efficient Training” ICML 2024.
>
> [3] Zhen Qin et al. “HGRN2: Gated Linear RNNs with State Expansion” https://arxiv.org/abs2404.07904
>
> [4] Tri Dao and Albert Gu. “Transformers are SSMs: Generalized Models and Efficient Algorithms Through Structured State Space Duality” ICML 2024.
>
> [5] Bo Peng et al. “Eagle and Finch: RWKV with Matrix-Valued States and Dynamic Recurrence” https://arxiv.org/abs/2404.05892
>
> [6] Maximilian Beck et al. “xLSTM: Extended Long Short-Term Memory” https://arxiv.org/abs/2405.04517
>
>
> > Initial states and length generalization
>
> We are very excited about passing states through the model so that forms of truncated back-propagation can be used to potentially learn on indefinitely long sequences. This functionality and other large improvements will be open-sourced to continue contributing to the community!

---

### Official Review · Reviewer_F6hZ · 2024-05-12

**Rating:** 6
**Confidence:** 5
**Ethics Flag:** 1

**Summary:**

This paper proposes Mamba, a linear-time sequence model with an intra-layer combination of Selective S4D, Short Convolution and Gated Linear Unit. The paper introduces input-dependent gating to SSM and provides a hardware-aware efficient implementation for the selective parallel scan operator. Mamba shows strong performance in language, DNA and audio modeling compared to models based on attention/RNN/convolution.

**Questions To Authors:**

1. How is the training wall time of Mamba compared to other models of similar sizes?
2. How is the performance of Mamba compared to the Llama architecture on downstream NLP tasks?
3. Is the short convolution module critical to Mamba's performance? What if we remove it?
4. It seems that in the Mamba official implementation, the $\mathbf{B}$ parameter in S6 is not using discretization, but rather a simple outer product. How does this discrepency affect the performance?  Does this indicate discretization is not neccessary for gated linear recurrent models? What about removing discretization for A? How is the discretization important generally?
5. Why is Mamba not length extrapolable in real-world language modeling?
6. How is the FLOPs calcuated in the scaling law experiments? Why using the Chinchilla protocol which is built for Transformers instead of fitting a new curve for Mamba?

**Reasons To Accept:**

1. The strong performance of the model on both synthetic and the real world tasks.
2. The proposed Selective State Space Models and its efficient implementation are significant contributions to the community of SSM and linear recurrence.

**Reasons To Reject:**

1. Lack of Ablation Study: Besides the proposed Selective S4D module, the Mamba layer also includes the Short Convolution module which can also be added to other linear recurrent models. It is not clear how Short Convolution contributes to the performance of Mamba.
2. Weak Baselines: The table 2 does not include the state-of-the-art transformer++ architecture that is used by Llama models, and the included baselines either have suboptimal architectural configurations or bad optimization/initialization recipes.
3. Lack of Training Efficiecy Measurement: The strong performance of Mamba can be due to the fact that it has far more layers than other models. The authors should show the training cost of Mamba when compared with other language models in Table 2.
4. The relationship to other input selection mechnism: While the authors claims that they "use selection to refer to the mechanistic **action** of a model to select or ignore inputs", Mamba does not make any discrete actions to select the input, but rather uses a soft gating techinque similar to Gated Recurrent Units. If the authors still want to use the terminology of selection, they should discuss its relationship to the works [1, 2, 3, 4] that have real discrete selections of inputs.
5. The length extrapolation experiments are only conducted on the synthetic setting. The synthetic induction head task is too simple and the authors make a strong claim of perfect length extrapolation. However, this doesn't even translate to the language modeling setting where Mamba is known to be not extrapolable as tested in [5] and [6]. I would suggets authors to (i) conduct experiments on a more challenging synthetic setting such as Multi-Query Associative Recall (MQAR), and (ii) avoid claiming the infinite extrapolation ability of Mamba.
6. The settings of the scaling law experiments are unclear and somewhat misleading. See question 6 below for more details. Also, in Figure 4, while increasing the sequence length from 2K to 8K, Mamba seems to have a worse scaling trend compared with Transformer++ when the training FLOPs is larger than $10^{20}$.

---
[1] Efficient Content-Based Sparse Attention with Routing Transformers (TACL 2020)

[2] Reformer: The Efficient Transformer (ICLR 2020)

[3] Switch Transformers: Scaling to Trillion Parameter Models with Simple and Efficient Sparsity (JMLR 2022)

[4] Sparse Modular Activation for Efficient Sequence Modeling (NeurIPS 2023)

[5] https://github.com/jzhang38/LongMamba

[6] Gated Linear Attention Transformers with Hardware-Efficient Training (ICML 2024)

---

> ### Author Rebuttal · Authors · 2024-05-31
>
> We thank the reviewer for the detailed feedback and insightful questions.
>
> 1. Ablation Study: The short conv is a simplification of H3’s “shift SSM”. RWKV has “token shift” (equivalent conv with width=2). The short conv does improve model quality (at 350M with 7B tokens, ppl 8.71 with conv and 8.97 without conv). The primary improvement comes from selectivity (Table 6).
> Extensive ablation experiments are included in Appendix E.3 (in Table 5, 6, 7, 8, 9).
>
> 2. Baselines:  We compared to Transformer++ in our scaling law study, showing that Mamba can match Transformer++ on perplexity. For table 2 (models up to 3B trained for 300B tokens), we compared to public models trained with similar numbers of tokens. Due to compute budget, we did not train our own Transformers. The fact that an SSM ( linear in seqlen) can match the strongest Transformer (quadratic in seqlen) and outperform other Transformer variants, which has gone through 7 years of intense tuning from the whole research community, is very encouraging to us.
>
> 3. Training Efficiency:  Mamba has the same number of layers as Transformers (e.g. Mamba 2.7B has 64 S6 layers, Pythia 2.8B has 32 MLP and 32 attn layers, same total params). Mamba’s training cost is comparable to that of Pythia (Pythia 2.8B takes 14240 A100 hours, Mamba-2.7B takes 14370).
>
> 4. Relationship to other input selection mechanisms:  We view this as analogous to “hard attention” (e.g. in Mnih et al., 2024 "Recurrent models of visual attention"), and “soft attention” (e.g. Bahdanau et al., 2015 “Neural machine translation …”), nowadays just called “attention” (e.g. by Vaswani et al., 2017).
>
> 5. Length extrapolation: We will make clear that our claims are with respect to specific empirical results in (1) synthetic tasks (2) audio (3) genomics. Follow-up works have further investigated long-context tasks, such as needle-in-a-haystack (https://github.com/jzhang38/LongMamba) and more realistic NLP tasks such as question-answering (Jamba), where Mamba (or variants) perform well.
>
> 6. Settings of the scaling law experiments:
> Full experiment settings are in Appendix F. For Transformer FLOPs: 6 * nparams * ntokens + attention FLOPs (12 * nlayers * hdim * ctx_len * ntokens). For Mamba: 6 * nparams * ntokens + scan FLOPs (27 * nlayers * hdim * state_dim * ntokens).
> The Chinchilla (20 tokens per params) protocol was originally tuned for Transformers, and we do not change this (1) due to compute budget (2) to be generous to the Transformer baselines.

---

### Official Review · Reviewer_hthE · 2024-05-14

**Rating:** 9
**Confidence:** 5
**Ethics Flag:** 1

**Summary:**

The authors integrate selective SSMs into a simplified end-to-end neural network architecture without attention or even MLP blocks (Mamba). Mamba enjoys fast inference (5× higher throughput than Transformers) and linear scaling in sequence length, and its
performance improves on real data up to million-length sequences. Mamba achieves state-of-the-art performance across
several modalities such as language, audio, and genomics. On language modeling, Mamba-3B model outperforms Transformers of the same size and matches Transformers twice its size, both in pretraining and downstream evaluation.

**Reasons To Accept:**

1. Mamba not only solves synthetic tasks such as copying and induction heads easily but also can extrapolate solutions indefinitely long (>1M tokens).
2. Mamba out-performs prior state-of-the-art models such as SaShiMi, Hyena, and Transformers on modeling audio waveforms and DNA sequences, both in pretraining quality and downstream metrics
3. Mamba is the first linear-time sequence model that truly achieves Transformer quality performance, both in pretraining perplexity and downstream evaluations.
4. Mamba is a hardware-aware algorithm. The resulting implementation is faster than previous methods.

Personally tested Mamba and it's amazing as the authors claimed.

**Reasons To Reject:**

Don't find a good reason to reject.

---

> ### Author Rebuttal · Authors · 2024-05-31
>
> We are very glad that the reviewer enjoyed the paper and model. Thanks for the encouragement!

---

### Official Review · Reviewer_YEh1 · 2024-05-15

**Rating:** 6
**Confidence:** 4
**Ethics Flag:** 1

**Summary:**

The paper proposes an improvement over state-space models that solve issues resulting from static nature of SSMs. The main idea is that the parameters of SSMs (e.g., CNN filters) are not input-dependent. The proposed method gives the model the ability to solve tasks that require "context-aware reasoning".

While this change sacrifices the efficiency of static SSMs, the paper also proposes a hardware kernel to speed up the models. The model demonstrates strong results on various modalities including text, DNA, audio etc.

Overall, the paper presents a very interesting model that has clear motivations, with solid executions. However, the experiments results, especially the language part look insufficient to show its superiority over transformers.

**Questions To Authors:**

How good can the model be parallelized and utilize machine compute?

Can you explain why only B, C needed to be context-aware and the static A can still be used?

The writing of the paper requires a good understanding of the precious S4 models, and I think it is worthwhile to add more context to the main paper instead of leaving them in the appendix. If space is an issue, I suggest the authors move the experiments of other modalities to the appendix given their low relevance to this conference.

**Reasons To Accept:**

* The method is a solid improvement over S4. The results on synthetic tasks clearly highlight the drawbacks of vanilla SSMs and validated the improvements.
* The efficient implementation proposed along with the model is quite interesting and seems useful to speed up other recurrent-style models.
* The experimental results cover a range of modalities, demonstrating the models' potential in diverse applications.

**Reasons To Reject:**

* Lack of results on long-context text tasks to justify that the models can actually replace transformers.
* It is unclear whether the comparison with Transformers is totally fair, e.g., training cost in terms of wall clock time even though the Flops might be smaller, if the learning rate schedule has been properly tuned for transformers.
* While the model has input-dependent behaviors, it still has a fixed size memory/hidden state of all previous context. This suggests that it might not work well in LLMs scenarios where the users can have different queries over the same context after the context has been encoded.

---

> ### Author Rebuttal · Authors · 2024-05-31
>
> Thanks for the feedback and questions!
>
> We would like to note that we view this work as re-invigorating interest into alternative architectures, and opening new directions in this space. The reviewer raises many interesting questions about other tasks/applications and potential efficiency improvements; while we could not address all of them in this first work, we are optimistic that much more progress and empirical validation will continue to be made.
>
> > Long-context tasks
>
> To our knowledge there are not standard and high-quality benchmarks that are widely agreed to measure long context in text. Thus our goal was to focus on more standard LM pretraining objectives, as well as other modalities (e.g. DNA) where long context matters.
> We note that follow-ups have further investigated long-context tasks, such as needle-in-a-haystack [LongMamba] and question-answering [Jamba], where Mamba (or variants) perform well.
>
> > Training efficiency
>
> We note that Mamba is much faster than Transformers in wall-clock time for longer sequences.
> However, as the reviewer observes, on a *FLOP-for-FLOP* basis, it is still less hardware-efficient than Transformers because Mamba does not utilize as many matrix multiplications, which GPUs are extremely specialized for.
>
> This is in fact an area of active research with very positive progress. For example, several follow-up works have been able to develop closely related models that leverage matrix multiplications and are much faster than Mamba [GLA, HGRN2, Mamba-2], comparable to Transformers in wall-clock time even for shorter sequences.
>
> > Fixed state size.
>
> Indeed, the tension between state size and recall ability is a fundamental tension of sequence models. We expect SSMs and Transformers to have different tradeoffs in inductive bias and capabilities. For example, SSMs naturally have more trouble with rote memorization [1], but have better ability to learn other types of formal languages [2]. In fact, we view SSMs and attention as complementary, and follow-up work has found that hybrid architectures are broadly very strong (e.g. Jamba, Zamba, Griffin).
>
> [1] https://arxiv.org/abs/2402.01032
> [2] https://arxiv.org/abs/2405.17394
>
> > Static A
>
> The discrete A dynamics $\bar{A}$ (which govern the SSM recurrence in eq. (2a)) depend on both $A$ and $\Delta$. Since the latter is context-aware, so is $\bar{A}$, which is the true recurrence. In early explorations, we found that additionally making $A$ input-dependent did not help performance.

---

> > ### Comment · Reviewer_YEh1 · 2024-06-04
> >
> > Thank you for the replies and for sharing interesting future directions! All of them make sense and I would recommend the authors to be clear about those limitations (in language tasks) in the paper itself. I think that would actually make this paper stronger -- instead of leaving an impression of overclaim, it's better to have a deep discussion about the difference, pros and cons of different models.
> >
> > As the authors suggested, this direction is still a work-in-progress and a couple of improvements are needed. But still, I think it's a worthy direction and it would be an interesting paper to be published in COLM.

---

### Decision · Program_Chairs · 2024-07-10

**Decision:**

Accept

**Comment:**

Mamba addresses key architectural and implementational problems in S4, creating a potentially viable alternative to the transformer family of architectures. While the jury may still be out on what is the optimal architecture for modeling long sequences, this is a significant advancement in sequence modeling with sub-quadratic complexity.

As reviewers YEh1 and F6hZ suggest, additional evaluations, ablations, and comparisons would have cemented the claims even more; I also acknowledge that this is very difficult to do in an academic setting. The paper presents sufficient results in my view, and as the authors and additional reviewers have noted, there is a growing amount of external evidence and adoption to support the paper's claims.